# EnerGS: Energy-Based Gaussian Splatting with Partial Geometric Priors

**Rui Song** [1 2]  **Tianhui Cai** [1]  **Markus Gross** [3]  **Yun Zhang** [1]  **Walter Zimmer** [1]  **Zhiyu Huang** [1]
**Olaf Wysocki** [2]  **Jiaqi Ma** [1]

## Abstract

3D Gaussian Splatting (3DGS) has been widely adopted for scene reconstruction, where training inherently constitutes a highly coupled and non-convex optimization problem. Recent works commonly incorporate geometric priors, such as Li-DAR measurements, either for initialization or as training constraints, with the goal of improving photometric reconstruction quality. However, in large-scale outdoor scenarios, such geometric supervision is often spatially incomplete and uneven, which limits its effectiveness as a reliable prior and can even be detrimental to the final reconstruction. To address this challenge, we model partially observable geometry as a continuous energy field induced by geometric evidence and propose EnerGS. Rather than enforcing geometry as a hard constraint, EnerGS provides a soft geometric guidance for the optimization of Gaussian primitives, allowing geometric information to steer the optimization process without directly restricting the solution space. Extensive experiments on large-scale outdoor scenes demonstrate that, under both sparse multi-view and monocular settings, EnerGS consistently improves photometric quality and geometric stability, while effectively mitigating overfitting during 3DGS training. The codebase is publicly available at: https://github.com/ucla-mobility/EnerGS.

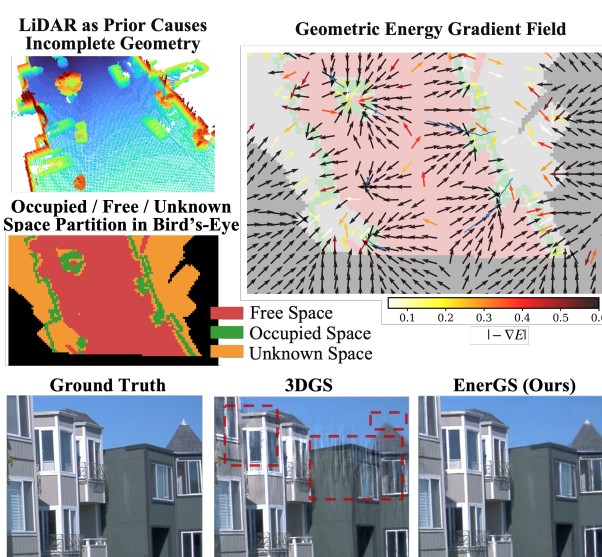

*Figure 1.* Accurate geometric priors can significantly improve Gaussian initialization and optimization (e.g., via point clouds from LiDAR). However, in large-scale outdoor scenes, such priors are often spatially incomplete. EnerGS addresses this limitation by partitioning space into occupied, free, and unknown regions and guiding the spatial distribution of Gaussians with a geometric energy field, enabling more stable novel-view rendering in regions unobserved by LiDAR than vanilla 3DGS.

## 1. Introduction

The field of novel view synthesis has witnessed a paradigm shift with the advent of 3D Gaussian Splatting (3DGS) [24, 17, 45, 4, 18, 48, 15, 50, 44, 38]. By representing scenes as a collection of anisotropic 3D Gaussians and utilizing a highly optimized differentiable rasterizer, 3DGS achieves real-time rendering speeds while maintaining photorealistic quality comparable to Neural Radiance Fields (NeRFs) [32]. While this explicit representation excels in bounded, object-centric scenarios with dense multi-view coverage, extending its efficacy to large-scale, unbounded environments, such as driving scenarios, presents distinct challenges. In such sparse-view regimes, the reconstruction problem becomes inherently ill-posed. Without sufficient constraints, the optimization often gravitates towards geometrically invalid local minima, spawning *floaters* or near-camera artifacts to overfit training views [46, 29].

To constrain this ill-posedness, a common strategy is to incorporate depth priors directly into the optimization objective [8, 41, 40, 26]. However, existing methods often treat sensor supervision uniformly, which may not fully account for the inherent discrepancy between modalities, i.e., geometric unobservability does not imply visual occlusion.

[1]University of California, Los Angeles, California, USA [2]University of Cambridge, Cambridge, UK [3]Technical University of Munich, Munich, Germany. Correspondence to: Rui Song <rruisong@ucla.edu>.

*Proceedings of the 43$^{rd}$ International Conference on Machine Learning*, Seoul, South Korea. PMLR 306, 2026. Copyright 2026 by the author(s).

In autonomous driving scenarios [37, 35], LiDAR sensors typically have a limited vertical Field-of-View (FoV) and sparsity, leaving vast regions (e.g., upper buildings, distant backgrounds) geometrically unobserved ($\Omega_{unk}$) yet clearly visible in camera images. Similar partial geometric observability is also common in other robotics settings, such as multi-agent perception and aerial scene completion [52, 12]. Applying rigid depth or smoothing priors globally can be suboptimal in these regions, as it risks suppressing valid structures that are strictly observable photometrically. Conversely, in regions where LiDAR provides definitive free space information, the absence of strict exclusion mechanisms can allow artifacts to accumulate.

In this work, we argue that robust reconstruction requires a principled partition of trust based on sensor observability. We propose Energy-Based Gaussian Splatting (EnerGS), a framework that reformulates 3DGS optimization as inference within a geometric energy field, as shown in Fig. 1. Our key insight is that the geometric constraint must be adaptive: it should be *rigid* in regions with active sensor coverage to enforce physical validity, but *compliant* in geometrically unobserved regions to stabilize primitives emerging from photometric densification. This flexibility is essential to bridge the gap between sensors: it allows the system to strictly reject floaters in verified free space while permitting the reconstruction of LiDAR-blind but camera-visible geometry.

To realize this, we construct a continuous geometric energy field that strictly partitions the domain into three distinct regimes. This field integrates a Welsch M-estimator as a volumetric attractor in both verified and potentially occupied regions, and a Boltzmann barrier as a strictly monotonic repulsor in certified free space. Crucially, within the geometrically unobserved regions (treated as potentially occupied), we instantiate this attractor with high variance, effectively modeling it as a weak prior. This design establishes a spatially adaptive potential landscape. Instead of applying a uniform regularization globally, the field enforces rigid physical constraints where sensor data is definitive, while imposing a soft, high-uncertainty prior in blind spots, thereby unifying deterministic sensor evidence and unobserved ambiguity into a single differentiable framework.

Our contributions are summarized as follows:

- We introduce an energy field that unifies uncertain-aware occupancy attraction (via a Welsch M-estimator) and free space exclusion (via a Boltzmann barrier) into a differentiable potential, explicitly handling LiDAR sparsity and noise.

- We propose an energy-driven decoupled optimization strategy that resolves the gradient conflict between geometry and appearance, theoretically guaranteeing

monotonic energy descent and the elimination of stable floaters in trusted free space.

- We demonstrate state-of-the-art performance on large-scale datasets (KITTI, Waymo), where our method significantly reduces geometric violations, while achieving superior rendering quality compared to existing regularized 3DGS approaches.

## 2. Related Work

**Efficient Radiance Fields and Gaussian Splatting.** NeRF [32] has introduced novel view synthesis by implicitly modeling scenes with Multilayer Perceptrons (MLPs). While capable of high-fidelity reconstruction, NeRFs [2, 3, 47, 14] suffer from prohibitive training and rendering costs. To address this, 3DGS [24] has introduced an explicit representation using anisotropic 3D Gaussians. By leveraging a tiled differentiable rasterizer, 3DGS achieves real-time rendering speeds and fast convergence [7, 31, 16, 51, 39, 42]. However, the explicit nature of 3DGS makes it prone to overfitting in unobserved regions. Unlike the continuous MLP field in NeRF, which has an inductive bias towards smoothness, discrete Gaussians can easily degenerate into "floaters" or needle-like artifacts in sparse-view settings, particularly in large-scale unbounded scenes where camera coverage is limited.

**Geometric Regularization in 3DGS.** To mitigate geometric artifacts, recent works have proposed various regularization strategies [5, 13, 22, 10]. Mip-Splatting [46] introduces 3D smoothing and 2D scale constraints to alleviate aliasing artifacts. Scaffold-GS [29] employs a hierarchical voxel anchor structure to stabilize primitive growth. Other approaches focus on surface alignment: 2DGS [19] flattens 3D Gaussians into 2D surfels to encourage surface smoothness , while DN-Splatter [36] incorporates depth normal consistency. While effective for bounded scenes or reducing high-frequency noise, these methods primarily rely on intrinsic photometric cues or local smoothness assumptions. They often lack a global mechanism to explicitly distinguish between empty space and occupied surfaces, leaving them vulnerable when photometric information is ambiguous or misleading.

**Multi-modal Fusion and LiDAR Priors.** Integrating geometric priors from depth sensors or LiDAR has been a longstanding strategy in 3D reconstruction. Early NeRF-based methods utilized sparse depth points to constrain the density field. In the context of 3DGS, multiple works incorporate depth or LiDAR observations as additional geometric constraints beyond image-only supervision. For example, GeoGaussian [28] and Taming-3DGS [30] initialize Gaussians from point clouds and apply densification constraints based on geometry. Beyond initialization, some works introduce depth- or LiDAR-based supervision during optimiza-

tion [20, 27, 25, 43, 6, 9, 49]. For example, LiGSM [33] leverages LiDAR observations to provide complementary geometric constraints, and LI-GS [23] incorporates LiDAR measurements for surface supervision during optimization.

However, most existing frameworks model geometric data as a set of *attraction targets* (e.g., minimizing $L_2$ distance to depth maps). This formulation has two limitations. First, it typically ignores the *free space evidence*—the empty volume traversed by LiDAR rays—which is crucial for removing floaters. Second, these methods typically optimize geometric loss ($\mathcal{L}_{\text{geom}}$) and photometric loss ($\mathcal{L}_{\text{photo}}$) simultaneously via a unified gradient descent. As discussed in Sec. 3.3, this *coupled* optimization often leads to gradient conflicts in occluded regions, where the noisy photometric gradient overrides the geometric signal. In contrast, our EnerGS framework treats LiDAR as a continuous energy field encompassing both attraction and repulsion, and employs a decoupled update rule to ensure geometric monotonicity.

## 3. Methodology

We present EnerGS, a framework that regularizes volumetric reconstruction by enforcing geometric priors derived from partially observed geometry information. We formulate the problem as a Maximum A Posteriori (MAP) estimation, decoupling the optimization of geometric attributes (position) from appearance attributes (radiance) to ensure robustness under partial observability.

### 3.1. Preliminaries: 3D Gaussian Splatting

We adopt the scene representation from vanilla 3DGS [24]. The scene is modeled as a set of $N$ anisotropic 3D Gaussians $\mathcal{G} = \{G_i\}_{i=1}^{N}$. Each primitive $G_i$ is defined by a mean position $\mu_i \in \mathbb{R}^3$ and a covariance matrix $\Sigma_i \in \mathbb{R}^{3 \times 3}$:

$$G_i(x) = \exp\left(-\frac{1}{2}(x - \mu_i)^\top \Sigma_i^{-1}(x - \mu_i)\right). \quad (1)$$

where $x \in \mathbb{R}^3$ is a 3D point at which a Gaussian can be evaluated. To render an image, 3D Gaussians are projected onto the 2D image plane via the viewing transformation $W \in \mathbb{R}^{3 \times 3}$ and the Jacobian $J \in \mathbb{R}^{2 \times 3}$ of the affine approximation. The resulting 2D covariance is $\Sigma_i' = JW\Sigma_i W^\top J^\top \in \mathbb{R}^{2 \times 2}$. The pixel color $C(u)$ for a pixel $u \in \mathbb{R}^2$ is computed via $\alpha$-blending of $K$ ordered primitives overlapping the pixel:

$$C(u) = \sum_{i=1}^{K} c_i \alpha_i \prod_{j=1}^{i-1}(1 - \alpha_j), \quad (2)$$

where the $K$ primitives are sorted by depth along the ray through $u$, $c_i$ represents view-dependent color (Spherical Harmonics), and $\alpha_i$ is the opacity derived from the 2D Gaussian evaluation.

Standard optimization updates all parameters $\Theta_i = \{\mu_i, \Sigma_i, \alpha_i, c_i\}$ by descending the gradient of the photometric loss $\mathcal{L}_{\text{photo}} = \lambda_1 \mathcal{L}_1 + \lambda_2 \mathcal{L}_{\text{D-SSIM}}$:

$$\Theta_i^{(t+1)} \leftarrow \Theta_i^{(t)} - \eta \frac{\partial \mathcal{L}_{\text{photo}}}{\partial \Theta_i}. \quad (3)$$

**Problem Statement.** In sparse-view or untextured regions, the partial derivative w.r.t. position, $\frac{\partial \mathcal{L}_{\text{photo}}}{\partial \mu_i}$, becomes ill-posed. This term is derived solely from rasterization errors and lacks 3D spatial awareness, often driving $\mu_i$ into empty space to overfit specific training views (floaters). We replace this unreliable term with a robust geometric gradient.

### 3.2. Probabilistic Geometric Field

We formally define the alignment of Gaussians with partially observed geometry priors, such as LiDAR measurements in outdoor scenarios, as the minimization of a geometric energy potential $E_{\text{geom}}$. Let $\mathcal{I} = \{I_v\}_v$ denote the set of images, and let $\mathcal{P}_{\text{LiDAR}} = \{p_k\}_k \subset \mathbb{R}^3$ denote a LiDAR point cloud. We treat the LiDAR measurements not just as a set of points, but as a source of a spatial probability density function $P_{\text{geom}}(x)$. Assuming the geometric prior is independent of the photometric appearance, our objective is to maximize the posterior:

$$P(\Theta | \mathcal{I}, \mathcal{P}_{\text{LiDAR}}) \propto \underbrace{P(\mathcal{I}|\Theta)}_{\text{Photometry}} \cdot \underbrace{P(\mu|\mathcal{P}_{\text{LiDAR}})}_{\text{Geometry}}. \quad (4)$$

This conditional independence assumption applies at the sensor-observation level: given the scene parameters $\Theta$, RGB and LiDAR observations are modeled as independent sensing processes.

We define the geometric energy as the negative log-likelihood of the geometric prior: $E_{\text{geom}}(\mu) = -\log P(\mu|\mathcal{P}_{\text{LiDAR}})$. Based on ray-tracing analysis of LiDAR data, we partition the domain $\Omega \subset \mathbb{R}^3$ into Occupied ($\Omega_{\text{occ}}$), Free ($\Omega_{\text{free}}$), and Unknown ($\Omega_{\text{unk}}$) regions, deriving specific energy potentials for each.

#### 3.2.1. ROBUST ATTRACTION IN OCCUPIED SPACE

For regions supported by LiDAR points $p_k \in \mathcal{P}_{\text{LiDAR}}$, we model the surface likelihood using a robust Welsch M-estimator [1]. Unlike $L_2$ minimization, which assumes a Gaussian distribution of errors on distances ($E \sim d^2$), the Welsch formulation assumes a bounded influence, preventing distant outliers from exerting excessive gradients. The energy potential $E_{\text{occ}} : \mathbb{R}^3 \to \mathbb{R}$ is defined as:

$$E_{\text{occ}}(\mu) = -w_{\text{occ}} \exp\left(-\frac{d_{\text{occ}}(\mu)^2}{2\sigma_{\text{occ}}^2}\right), \quad (5)$$

where $d_{\text{occ}}(\mu) = \min_{p \in \mathcal{P}_{\text{LiDAR}}} \|\mu - p\|$. The force (negative partial derivative) acting on a Gaussian at $\mu$ is:

$$-\frac{\partial E_{\text{occ}}}{\partial \mu} = \frac{1}{\sigma_{\text{occ}}^2} E_{\text{occ}}(\mu) \cdot (\mu - p_{\text{nn}}), \qquad (6)$$

where $p_{\text{nn}}$ is the nearest LiDAR point. This creates a local vector field that pulls primitives towards the surface, with a magnitude that asymptotically decreases to zero with increasing distance.

### 3.2.2. BOLTZMANN BARRIER IN FREE SPACE

In certified free space $\Omega_{\text{free}}$, the existence of a primitive is a physical violation. We model the probability of a "valid" surface existing at depth $s$ inside free space using a Boltzmann distribution $P(s) \propto \exp(-s/\tau)$. The associated energy is linear with respect to the penetration depth $d_{\text{trust}}(\mu)$. To ensure $C^1$ continuity for optimization, we employ a Softplus barrier:

$$E_{\text{free}}(\mu) = \lambda_{\text{free}} \cdot \text{softplus}\left(\frac{d_{\text{trust}}(\mu) - \delta}{\tau}\right). \qquad (7)$$

The partial derivative exerted by this field is:

$$\frac{\partial E_{\text{free}}}{\partial \mu} = \frac{\lambda_{\text{free}}}{\tau} \sigma\left(\frac{d_{\text{trust}} - \delta}{\tau}\right) \frac{\partial d_{\text{trust}}}{\partial \mu}. \qquad (8)$$

Here, $\sigma(\cdot)$ is the sigmoid function, and $\frac{\partial d_{\text{trust}}}{\partial \mu}$ represents the unit normal vector pointing towards the valid region boundary. This term applies a monotonic repulsive force that pushes primitives out of free space along the steepest descent direction.

### 3.2.3. WEAK PRIOR IN UNKNOWN SPACE

In unobserved regions $\Omega_{\text{unk}}$, we apply a weak regularization to prevent unbounded drift while allowing photometric optimization to dominate. We define a broad, low-amplitude potential:

$$E_{\text{unk}}(\mu) = -w_{\text{unk}} \exp\left(-\frac{d_{\text{unk}}(\mu)^2}{2\sigma_{\text{unk}}^2}\right). \qquad (9)$$

Crucially, we set $w_{\text{unk}} \ll w_{\text{occ}}$. This ensures that $|\frac{\partial E_{\text{unk}}}{\partial \mu}| \approx 0$ unless the primitive is extremely close to the prior center, effectively serving as a weak regularizer.

### 3.3. Optimization via Gradient Decoupling

Directly minimizing $\mathcal{L}_{\text{total}} = \mathcal{L}_{\text{photo}} + \lambda E_{\text{geom}}$ is suboptimal because the noisy photometric gradient $\frac{\partial \mathcal{L}_{\text{photo}}}{\partial \mu}$ often contradicts the geometric gradient $\frac{\partial E_{\text{geom}}}{\partial \mu}$, leading to local minima.

We propose a decoupled update rule. We explicitly block the flow of photometric gradients to the mean position $\mu$,

while allowing them to update covariance and appearance. The update rules for iteration $t$ are:

$$\mu^{(t+1)} \leftarrow \mu^{(t)} - \eta_\mu \left.\frac{\partial E_{\text{geom}}}{\partial \mu}\right|_{\mu^{(t)}}, \qquad (10)$$

$$\{\Sigma, \alpha, c\}^{(t+1)} \leftarrow \{\Sigma, \alpha, c\}^{(t)} - \eta_{\text{app}} \frac{\partial \mathcal{L}_{\text{photo}}}{\partial \{\Sigma, \alpha, c\}}. \qquad (11)$$

By enforcing Eq. (10), we guarantee that the spatial distribution of Gaussians evolves monotonically downwards on the geometric energy landscape. Eq. (11) simultaneously ensures that given these physically plausible positions, the Gaussians deform ($\Sigma$) and change color ($c$) to maximize photometric consistency. This separation effectively resolves the conflict between imperfect geometry and view-dependent appearance.

### 3.4. Discrete Pruning as Boundary Enforcement

While the continuous energy field $E_{\text{free}}$ exerts a repulsive force, extremely high photometric error can trigger aggressive densification, spawning clusters of artifacts deep in free space. If the repulsive velocity is too high, it may destabilize the rasterizer; if too low, artifacts persist. To mitigate this, we employ a *discrete hard pruning* strategy as a secondary barrier. Every $T_{\text{prune}}$ iterations, we verify the spatial state of all primitives:

$$\mathcal{G} \leftarrow \mathcal{G} \setminus \{G_i \mid d_{\text{trust}}(\mu_i) > \tau_{\text{margin}}\}. \qquad (12)$$

This acts as an additional pruning criterion for primitives that fail to escape free space within a time window, preventing the accumulation of floaters and maintaining training stability.

### 3.5. Complexity and Implementation Efficiency

We implement the geometric prior as a discretized voxel grid of resolution $V^3$ to decouple geometric queries from the expensive rendering pipeline. In the initialization step, we compute the Euclidean Distance Transform [11] for the LiDAR point cloud and derive the gradient field $\nabla E_{\text{geom}}$ via central differences. This $\mathcal{O}(V^3)$ operation is performed once, avoiding expensive runtime differentiation and ensuring that complex geometric constraints are reduced to simple lookups.

During training, the geometric regularization operates with linear complexity relative to the number of primitives $N$. At each iteration, we retrieve the gradient vector for every Gaussian via trilinear interpolation and apply the positional update $\mu \leftarrow \mu - \eta_\mu \nabla E_{\text{geom}}$. Both the field query and the pruning check $d_{\text{trust}}(\mu) > \tau_{\text{margin}}$ are vectorized $\mathcal{O}(N)$ operations that bypass the differentiable rasterizer entirely.

Consequently, the computational bottleneck remains the vanilla 3DGS sorting ($\mathcal{O}(N \log N)$) and pixel blending stages. Experimentally, our geometric module incurs negligible overhead, allowing the framework to maintain the training efficiency characteristic of 3D Gaussian Splatting.

# 4. Theoretical Analysis

In this section, we analyze the convergence properties of the proposed framework. We formulate the optimization as a dynamical system driven by a composite vector field and prove that the design of the geometric energy guarantees both the rejection of invalid solutions and the preservation of unobserved geometry.

## 4.1. Problem Formulation and Assumptions

We define the problem based on the properties of the solution space and the spatial partition of the priors.

**Assumption 1 (Manifold of Photometric Ambiguity).** Let $\mathcal{L}_{\text{photo}}(\Theta)$ be the photometric loss. In sparse-view regimes, the solution manifold $\mathcal{S} = \{\Theta \mid \frac{\partial \mathcal{L}_{\text{photo}}}{\partial \Theta} = 0\}$ contains a subset of degenerate solutions $\mathcal{S}_{\text{deg}} \subset \mathcal{S}$. A solution $\Theta \in \mathcal{S}_{\text{deg}}$ is characterized by minimal training residuals despite significant positional deviation from the ground truth surface.

**Assumption 2 (Partition of Observability).** The spatial domain $\Omega \subset \mathbb{R}^3$ is strictly partitioned into two disjoint sets based on sensor observability: $\Omega = \Omega_{\text{trust}} \cup \Omega_{\text{unk}}$.

- **Trusted Domain ($\Omega_{\text{trust}}$):** The subset where the sensor provides definitive occupancy information (either occupied or free).

- **Unobserved Domain ($\Omega_{\text{unk}}$):** The complement subset where no sensor readings exist. This domain is modeled as a region with high geometric uncertainty.

**Assumption 3 (Variance Hierarchy).** The geometric energy potentials are parameterized such that the spatial variance in the unobserved domain ($\sigma_{\text{unk}}$) is strictly larger than the sensor noise floor ($\sigma_{\text{sensor}}$) used in the trusted domain. Specifically, we assume the regime where $\sigma_{\text{unk}} \gg \sigma_{\text{sensor}}$.

## 4.2. Exclusion of Degenerate Solutions

We first prove that degenerate solutions (floaters) cannot persist in the trusted free space, regardless of their photometric consistency.

**Theorem 1 (Instability of Degenerate Points).** *Let $\mu$ be the position of a Gaussian primitive. If $\mu$ lies within the trusted free space $\Omega_{\text{free}} \subset \Omega_{\text{trust}}$, it cannot be a stable stationary point of the decoupled update rule, even if $\mu \in \mathcal{S}_{\text{deg}}$.*

**Proof.** The decoupled position update is governed solely by the negative partial derivative of the geometric energy. For a point in $\Omega_{\text{free}}$, the energy is defined as $E_{\text{free}}(\mu) = \phi(d_{\mathcal{T}}(\mu))$, where $d_{\mathcal{T}}$ is the distance to the trusted boundary. The magnitude of the update vector $\mathbf{v}$ is:

$$\|\mathbf{v}\| \propto \|\frac{\partial E_{\text{free}}}{\partial \mu}\| = \|\phi'(d_{\mathcal{T}}) \frac{\partial d_{\mathcal{T}}}{\partial \mu}\|. \tag{13}$$

Given that $\phi$ is the monotonically increasing Softplus function ($\phi'(s) > 0$) and the gradient of the Euclidean distance transform is non-zero almost everywhere ($\|\frac{\partial d_{\mathcal{T}}}{\partial \mu}\| = 1$), it follows that $\|\mathbf{v}\| > 0$ for all $\mu$ where $d_{\mathcal{T}}(\mu) > 0$. Therefore, a primitive in $\Omega_{\text{free}}$ is subject to a persistent non-zero force field. It must migrate along the trajectory defined by $-\frac{\partial d_{\mathcal{T}}}{\partial \mu}$ until exiting $\Omega_{\text{free}}$, thereby ensuring $\mathcal{S}_{\text{deg}} \cap \Omega_{\text{free}} = \emptyset$ in the convergence limit. $\square$

## 4.3. Optimization Stability

We analyze the smoothness of the optimization trajectory by examining the Lipschitz properties of the driving force.

**Theorem 2 (Regularity of the Update Field).** *The geometric energy field $E_{\text{geom}}$ induces a gradient field with a bounded Lipschitz constant, regularizing the optimization trajectory against high-frequency photometric noise.*

**Proof.** Let the update function be $\mathcal{F}(\mu) = -\frac{\partial E_{\text{geom}}}{\partial \mu}$.

1. **Photometric Field:** The gradient derived from rasterization, $\frac{\partial \mathcal{L}_{\text{photo}}}{\partial \mu}$, contains discontinuities due to visibility jumps (occlusions) and sampling aliasing. Its local Lipschitz constant is unbounded ($L \to \infty$), causing oscillatory behavior.

2. **Geometric Field:** The energy $E_{\text{geom}}$ is constructed via convolution with smooth kernels (Gaussian) or smoothed distance functions. For $E_{\text{free}}$, the Hessian $\|\frac{\partial^2 E}{\partial \mu^2}\|$ is bounded by the smoothing parameter $\tau$. For $E_{\text{occ}}$, the Gaussian kernel is $C^\infty$ continuous.

Consequently, $\mathcal{F}(\mu)$ is Lipschitz continuous with a finite constant $L_{\text{geom}}$. By the Descent Lemma, updating positions along a vector field with finite Lipschitz constant guarantees monotonic energy reduction for a suitable step size, ensuring training stability. $\square$

## 4.4. Permissiveness via Asymptotic Variance Analysis

Finally, we demonstrate that the "Unknown" region naturally permits reconstruction driven by photometry, without requiring explicit heuristic switching. This is derived from the asymptotic behavior of the Welsch estimator.

**Proposition 3 (Vanishing Geometric Constraint).** *In the unobserved domain $\Omega_{\text{unk}}$, the magnitude of the geometric*

*gradient converges to zero as the prior variance $\sigma_{\text{unk}}$ increases. This implies that for sufficiently large $\sigma_{\text{unk}}$, the optimization becomes dominated by photometric density control.*

**Proof.** Consider the gradient magnitude of the energy potential $E(\mu) \propto -\exp(-\|\mu - p\|^2 / 2\sigma^2)$ used in the unobserved domain. The force magnitude $F = \|\frac{\partial E}{\partial \mu}\|$ is given by:

$$F(d, \sigma) = \frac{w \cdot d}{\sigma^2} \exp\left(-\frac{d^2}{2\sigma^2}\right), \qquad (14)$$

where $d = \|\mu - p\|$. We analyze the behavior of this force with respect to the uncertainty parameter $\sigma$. For any finite distance $d$, taking the limit $\sigma \to \infty$:

$$\lim_{\sigma \to \infty} F(d, \sigma) = \lim_{\sigma \to \infty} \frac{wd}{\sigma^2} \cdot \underbrace{\exp\left(-\frac{d^2}{2\sigma^2}\right)}_{\to 1} = 0. \quad (15)$$

The term $\frac{1}{\sigma^2}$ dominates the decay.

In $\Omega_{\text{unk}}$, where we model the prior with high variance (Assumption 3), the geometric update vector $\mathbf{v}_{geom} \to \mathbf{0}$. While the position update stagnates under geometric forces, the structural update (densification driven by $\frac{\partial \mathcal{L}_{\text{photo}}}{\partial \mu}$) remains active. Since the geometric penalty is negligible, primitives spawned by photometric error are retained. This mathematically justifies the system's ability to reconstruct geometry in blind spots (e.g., occlusion or far-field) solely through multi-view consistency. □

## 5. Experiments

### 5.1. Experimental Setup

We conduct a comprehensive evaluation on large-scale autonomous driving datasets to assess the efficacy of EnerGS in handling partially observable geometry. Beyond improvements in standard evaluation metrics, our primary objective is to validate that our proposed energy formulation successfully resolves the ill-posedness inherent in sparse LiDAR supervision. We show that by enforcing our derived geometric constraints, EnerGS eliminates free space artifacts while faithfully reconstructing unobserved yet visible structures, resulting in robust photometric and geometric performance.

**Datasets and Baselines.** We conduct our evaluation on the KITTI [37] and Waymo Open Dataset [35], selecting sequences characterized by complex occlusions and unbounded backgrounds. Our study focuses exclusively on static scenes, and consequently, the evaluation excludes all dynamic objects. We compare our method against the vanilla 3DGS [24] and a suite of recent state-of-the-art approaches targeting geometric regularization or anti-aliasing, including 2DGS [19], Taming-3DGS [30], GeoGaussian [28], Mip-Splatting [46], Scaffold-GS [29]. For

a fair comparison, we initialize all methods with identical point clouds whenever applicable. All methods are trained for 30,000 iterations. For each scene, test frames for novel view synthesis evaluation are sampled every fourth frame.

**Metrics.** To ensure a consistent comparison, we evaluate all methods using a unified set of photometric, geometric, and efficiency metrics across both datasets. Photometric quality is measured using PSNR and SSIM on held-out test views, assessing reconstruction fidelity and structural consistency. Geometric quality is evaluated using a common voxelized occupancy field derived from LiDAR observations. Specifically, *OccCov* measures the coverage of occupied space by reconstructed Gaussian primitives, *Leak* penalizes violations in known free space regions, *Margin* captures the spatial separation between occupied and free regions, and *Thick* measures the effective thickness of reconstructed surfaces, where thinner surfaces indicate sharper and more physically plausible geometry. Efficiency is quantified by the total number of Gaussian primitives (#G), reflecting the compactness of the representation. All geometric metrics are computed with identical filtering and evaluation criteria across methods, ensuring that observed differences stem from reconstruction quality rather than evaluation bias.

### 5.2. Quantitative Analysis

Table 1 shows that our method consistently achieves stronger geometric performance on both KITTI and Waymo datasets while maintaining competitive photometric quality. On KITTI, it attains the highest PSNR and OccCov together with the lowest Leak score, indicating improved alignment with occupied regions and fewer free space violations. On Waymo, the advantages become more pronounced under partial observability, where our method achieves the best Leak and Margin scores and the highest PSNR, demonstrating clearer separation between occupied and free space. Importantly, these gains are obtained with a comparable number of Gaussians, highlighting a more compact and physically consistent reconstruction. We further compare with the LiDAR-guided SplatAD [16] in Tab. 6, where EnerGS substantially improves KITTI performance by reducing Leak from 22.1 to 7.0 and increasing PSNR from 17.14 to 18.32, while remaining comparable on denser Waymo scenes.

### 5.3. Qualitative Results

Fig. 2 compares novel view synthesis results of state-of-the-art baselines and our EnerGS across different scenes. Existing methods typically incorporate LiDAR as a geometric prior only at initialization or implicitly assume geometric completeness, which causes them to fail in regions beyond LiDAR coverage, such as rooftops, sky, tree canopies, and thin structures like poles. Under sparse-view conditions, the absence of reliable geometric constraints leads to local

*Table 1.* **Quantitative Comparison on KITTI and Waymo Open Dataset.** We report Photometry (PSNR, SSIM), Geometry (Leak, OccCov, Margin, Thick), and Efficiency (#Gauss). **Bold**, underlined, and *italicized* values indicate the first, second, and third best results, respectively. "–" indicates missing data.

| Method | KITTI | | | | | | | Waymo Open Dataset | | | | | | |
| --- | --- | --- | --- | --- | --- | --- | --- | --- | --- | --- | --- | --- | --- | --- |
| | Photometry | | Geometry | | | | #G (M)↓ | Photometry | | Geometry | | | | #G (M)↓ |
| | PSNR↑ | SSIM↑ | Leak↓ | OccCov↑ | Margin↑ | Thick↓ | | PSNR↑ | SSIM↑ | Leak↓ | OccCov↑ | Margin↑ | Thick↓ | |
| 3DGS ToG 23 | 15.01 | 0.938 | 12.5 | 16.1 | 0.22 | 1.39 | 1.51 | *24.12* | 0.920 | 0.3 | 13.5 | 3.18 | 0.71 | 2.01 |
| 2DGS SIGGRAPH 24 | *15.57* | 0.897 | *12.1* | *19.8* | *0.20* | 1.40 | **0.71** | 23.61 | 0.905 | 0.6 | 18.4 | 3.20 | 0.73 | **0.74** |
| Taming-3DGS SIGGRAPH 24 | 15.73 | *0.929* | 12.8 | *19.8* | 0.22 | **1.37** | 1.75 | 24.07 | *0.918* | 0.3 | 15.4 | 3.20 | 0.72 | 2.26 |
| GeoGaussian ECCV 24 | 14.99 | 0.890 | 15.2 | 19.3 | 0.16 | 1.42 | *0.84* | 23.26 | 0.916 | 0.6 | **22.6** | 3.09 | **0.70** | 1.02 |
| Mip-Splatting CVPR 24 | 13.91 | **0.950** | 15.7 | 21.1 | 0.04 | **1.37** | 1.78 | 24.18 | **0.957** | 0.4 | 19.0 | 3.04 | 0.73 | 2.46 |
| Scaffold-GS SIGGRAPH 24 | 13.97 | 0.319 | 11.3 | 19.4 | **1.15** | **1.37** | 0.77 | 21.23 | 0.674 | 0.3 | *19.3* | 2.59 | 0.71 | *1.10* |
| **Ours** | **16.80** | 0.841 | **9.2** | **25.8** | 0.18 | 1.47 | 1.20 | **24.81** | *0.918* | **0.2** | 20.7 | **3.33** | 0.76 | 1.57 |

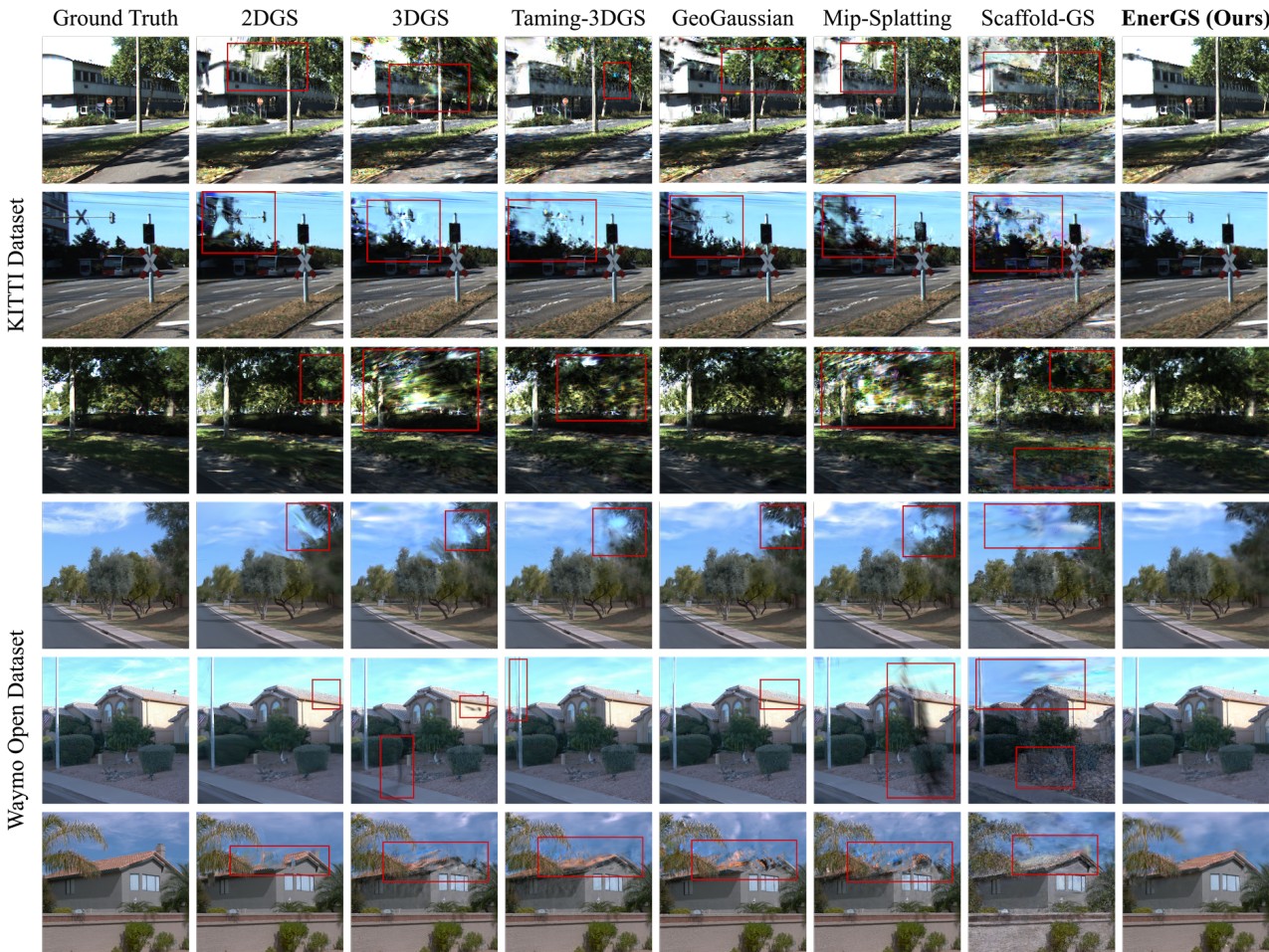

*Figure 2.* **Visual Comparison on KITTI and Waymo Open Dataset.** Our EnerGS achieves superior novel view synthesis, particularly in geometrically unobserved regions (e.g., upper structures beyond LiDAR coverage). Our method renders significantly finer details in these areas compared to baselines, aligning with our theoretical expectation that the adaptive energy field facilitates robust reconstruction in sensor blind spots.

photometric overfitting, driving Gaussian primitives into physically invalid positions. While such misalignment may not be apparent in training views, it manifests as floaters and structural distortions in novel viewpoints, degrading

the final photometric quality. In contrast, EnerGS explicitly models geometric uncertainty and applies weak regularization in unobserved regions, allowing photometric cues to guide reconstruction without enforcing incorrect geometry,

*Table 2.* **Ablation Study on KITTI and Waymo Open Dataset.** All values report differences ($\Delta$) relative to the full EnerGS model. Positive $\Delta$ indicates an increase compared to the baseline.

| Variant | KITTI | | | | | | | Waymo Open Dataset | | | | | | |
|---|---|---|---|---|---|---|---|---|---|---|---|---|---|---|
| | Photometry | | Geometry | | | | | Photometry | | Geometry | | | | |
| | $\Delta$PSNR↑ | $\Delta$SSIM↑ | $\Delta$Leak↓ | $\Delta$OccCov↑ | $\Delta$Margin↑ | $\Delta$Thick↓ | $\Delta$#G↓ | $\Delta$PSNR↑ | $\Delta$SSIM↑ | $\Delta$Leak↓ | $\Delta$OccCov↑ | $\Delta$Margin↑ | $\Delta$Thick↓ | $\Delta$#G↓ |
| Full EnerGS | 0.00 | 0.000 | 0.0 | 0.0 | 0.00 | 0.00 | 0.00 | 0.00 | 0.000 | 0.0 | 0.0 | 0.00 | 0.00 | 0.00 |
| Weak $\mathcal{L}_{unk}$ | -0.14 | +0.015 | +0.2 | -1.5 | +0.11 | +0.01 | -0.01 | -3.45 | +0.062 | +0.0 | -1.4 | -0.01 | -0.00 | +0.01 |
| w/o $\mathcal{L}_{unk}$ | -0.07 | +0.015 | +0.4 | -1.9 | +0.10 | +0.05 | +0.01 | -3.62 | +0.063 | -0.0 | -1.2 | -0.01 | +0.01 | +0.02 |
| w/o $\mathcal{L}_{free}$ | -0.30 | +0.012 | +0.0 | -0.6 | +0.11 | +0.01 | +0.00 | -3.64 | -0.065 | +0.0 | -0.3 | -0.01 | +0.01 | +0.00 |
| w/o $\mathcal{L}_{occ}$ | -0.67 | +0.036 | -1.8 | -4.5 | -0.05 | +0.06 | -0.09 | -3.31 | +0.064 | +0.1 | -0.6 | -0.06 | +0.00 | +0.02 |
| w/o Welsch (L2 Loss) | -0.29 | +0.005 | +0.6 | -1.6 | +0.12 | +0.10 | -0.02 | -3.60 | -0.065 | -0.0 | +3.2 | +0.02 | +0.01 | +0.01 |
| w/o Decoupling | -0.56 | +0.062 | -2.8 | -4.0 | +0.14 | +0.09 | +0.30 | -3.20 | +0.073 | -0.0 | -1.3 | +0.03 | -0.00 | +0.06 |

thereby producing more stable and visually consistent novel view synthesis across viewpoints. This visual evidence supports Theorem 1, confirming that the decoupled update rule effectively precludes the existence of stable degenerate solutions in $\Omega_{\text{free}}$.

### 5.4. Ablation Studies

The ablation results in Table 2 show that removing geometric energy terms or disabling the decoupled optimization leads to consistent degradation in both photometric quality and geometric consistency across KITTI and Waymo. In particular, $E_{\text{occ}}$ is essential for concentrating Gaussian primitives near surface regions in $\Omega_{\text{occ}}$, while $E_{\text{free}}$ and $E_{\text{unk}}$ regulate their behavior in $\Omega_{\text{free}}$ and $\Omega_{\text{unk}}$, respectively. Decoupled optimization further stabilizes the spatial evolution of Gaussian positions under partial observability.

Several ablation variants show reduced leakage ratios and increased margins while occupied coverage and surface alignment deteriorate. These effects arise from the spatial redistribution of Gaussian primitives when geometric constraints are weakened. Without $E_{\text{occ}}$ or decoupled optimization, Gaussians no longer concentrate near the Occ-Free interface and instead disperse within $\Omega_{\text{unk}}$. Consequently, fewer primitives align with occupied space, leading to lower OccCov. At the same time, many Gaussians remain farther from $\Omega_{\text{free}}$ boundaries, which reduces the fraction of UNK primitives intruding into free space and lowers Leak%. The same redistribution causes Gaussians to retreat from boundary regions, increasing the average Occ-Free separation measured by Margin, even though the geometry becomes less surface aligned.

### 5.5. Training Generalization Comparison

We further investigate the training dynamics to verify our stability claims. Fig. 3 shows the generalization gap between Train and Test PSNR across different baseline methods. While several baselines exhibit a large gap, indicating overfitting that degrades novel-view synthesis, our EnerGS

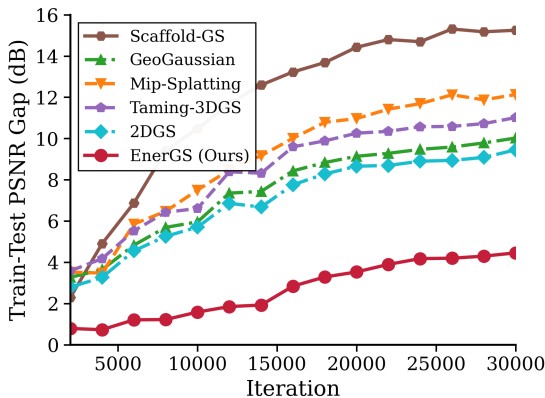

*Figure 3.* **The Gap between Train and Test PSNR with Training Iteration.** Our EnerGS consistently maintains a smaller train–test gap throughout training, indicating that our method encourages the model to learn multi-view consistent geometry rather than viewpoint memorization and ultimately achieves the best performance, as reported in Tab. 1.

consistently maintains a smaller gap throughout training. This suggests a more stable optimization process, where geometric priors encourage the learning of valid, multi-view consistent geometry rather than memorizing specific viewpoints, thereby improving generalization to unseen views under sparse-view training settings.

## 6. Conclusion

We propose EnerGS, an energy-based framework for 3D Gaussian Splatting that explicitly models partial and uneven geometric observability in large-scale outdoor scenes. By formulating LiDAR supervision as a spatially adaptive geometric energy field and decoupling Gaussian position updates from photometric optimization, EnerGS enforces strict physical validity in trusted free space while preserving flexibility in geometrically unobserved but visually supported regions. Our theoretical analysis characterizes the resulting optimization dynamics. It shows that degenerate solutions in free space cannot form stable equilibria and that the geometric update field is well-conditioned. In LiDAR-unobserved

regions, geometric constraints are automatically weakened, while photometric consistency continues to drive Gaussian densification. Extensive experiments on KITTI and Waymo closely match these theoretical predictions, demonstrating effective removal of free space artifacts, improved optimization stability, and reliable reconstruction in LiDAR blind spots, all with negligible computational overhead.

**Limitations.** EnerGS focuses on static large-scale scene reconstruction and does not explicitly model dynamic objects. In real-world autonomous driving scenes, moving vehicles, pedestrians, and cyclists may violate the static free-space and occupancy assumptions used to construct the geometric energy field. Although EnerGS can be integrated with foreground-background or dynamic Gaussian Splatting pipelines, handling dynamic agents within a unified energy formulation remains an important direction.

## Acknowledgments

This work was supported by the Federal Highway Administration Center of Excellence on New Mobility and Automated Vehicles, and by the National Science Foundation under Award No. 2346267, POSE: Phase II - DriveX: An Open Source Ecosystem for Automated Driving and Intelligent Transportation Research.

## Impact Statement

This work advances 3D Gaussian Splatting for large-scale outdoor scene reconstruction by addressing incomplete and uneven geometric supervision. EnerGS may support more reliable reconstruction in autonomous driving, robotic perception, and simulation-based evaluation, especially under sparse or partial LiDAR observations. At the same time, reconstruction systems for safety-relevant environments can pose practical and ethical risks, including inaccurate geometry, missing objects, or overconfident downstream interpretation. Although spatially adaptive geometric guidance helps reduce free-space artifacts and improve physical plausibility, the system remains subject to sensing, modeling, and optimization failures. Independent validation and human oversight are therefore necessary in safety-critical settings.

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

# Appendix

## A. Empirical Validation of Theoretical Analysis

### A.1. Empirical Validation of Theorem 1.

To empirically validate the theoretical properties defined in Theorem 1, we conducted a controlled Random Initialization Experiment. We explicitly initialized 500 Gaussian primitives within the free space and monitored their evolution. As illustrated in Fig. 4, vanilla 3DGS lacks the necessary constraints to reject invalid geometry, resulting in 315 out of 500 Gaussians remaining trapped in the empty region. In contrast, consistent with the definition in Theorem 1, EnerGS successfully expelled all initialized primitives from the free space. This result serves as direct numerical evidence that our proposed energy score effectively enforces the geometric constraints derived in our theoretical analysis.

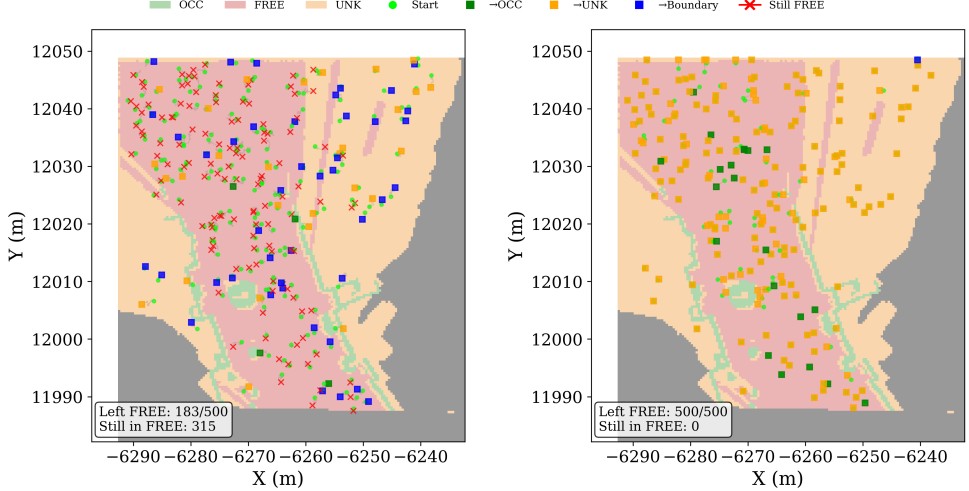

*Figure 4.* Random Initialization Experiment. We randomly initialize 50,000 Gaussian primitives (with 500 of them tracked and recorded) in free space and optimize them using vanilla 3D Gaussian Splatting (left) and EnerGS (right), respectively. The figure illustrates the initial and final positions of these Gaussians. Note that this visualization shows a slice at the camera height; therefore, some Gaussians may appear to lie within the free space region in the XY plane (in world coordinate system). However, when marked in green or orange, they indicate that the Gaussians have already exited the free space region in 3D space.

### A.2. Empirical Validation of Theorem 2.

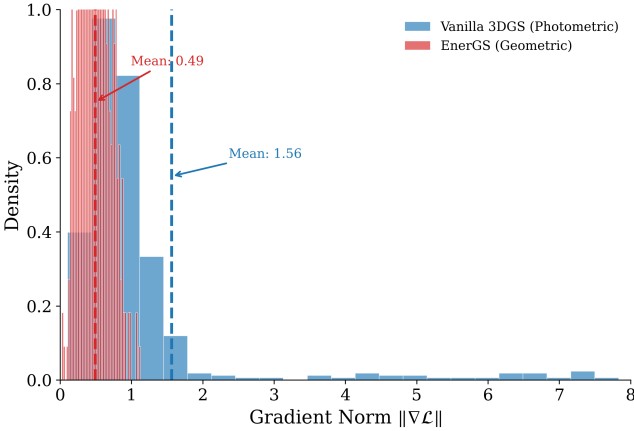

*Figure 5.* Comparison of Gradient Norm between Vanilla 3DGS and EnerGS.

We provide experimental evidence supporting the regularity properties of our geometric energy field. Fig 5 compares the gradient norm distribution between baseline photometric gradients and our geometric gradients. The photometric gradient (blue) exhibits a heavy-tailed distribution extending beyond $\|\nabla\mathcal{L}\| > 6$, confirming the presence of unbounded gradient spikes caused by visibility discontinuities as predicted by our analysis. In contrast, the geometric gradient (red) shows a concentrated distribution with bounded support, with mean gradient norm reduced from 1.56 to 0.49. This empirically validates that $E_{\text{geom}}$ possesses a finite Lipschitz constant $L_{\text{geom}}$.

### A.3. Empirical Validation of Proposition 3.

As illustrated in Fig. 6, we compare the two strategies within LiDAR blind spots (e.g., rooftops and sky) on the Waymo sequence. The Strict strategy aggressively prunes all primitives in UNK regions; while this eliminates potential noise, it inadvertently discards structures that are visible in images yet missed by LiDAR, resulting in significant missing geometry and background leakage (visible as "blue holes"). In contrast, EnerGS successfully reconstructs complete rooftop profiles by leveraging photometric cues. This result visually validates Proposition 3: due to the decay of the geometric gradient in UNK regions (i.e., asymptotic stability), Gaussian primitives within these blind spots are shielded from spurious geometric forces. This "positional stability" allows them to be safely driven solely by photometric loss, enabling the precise recovery of missing scene details without introducing geometric artifacts.

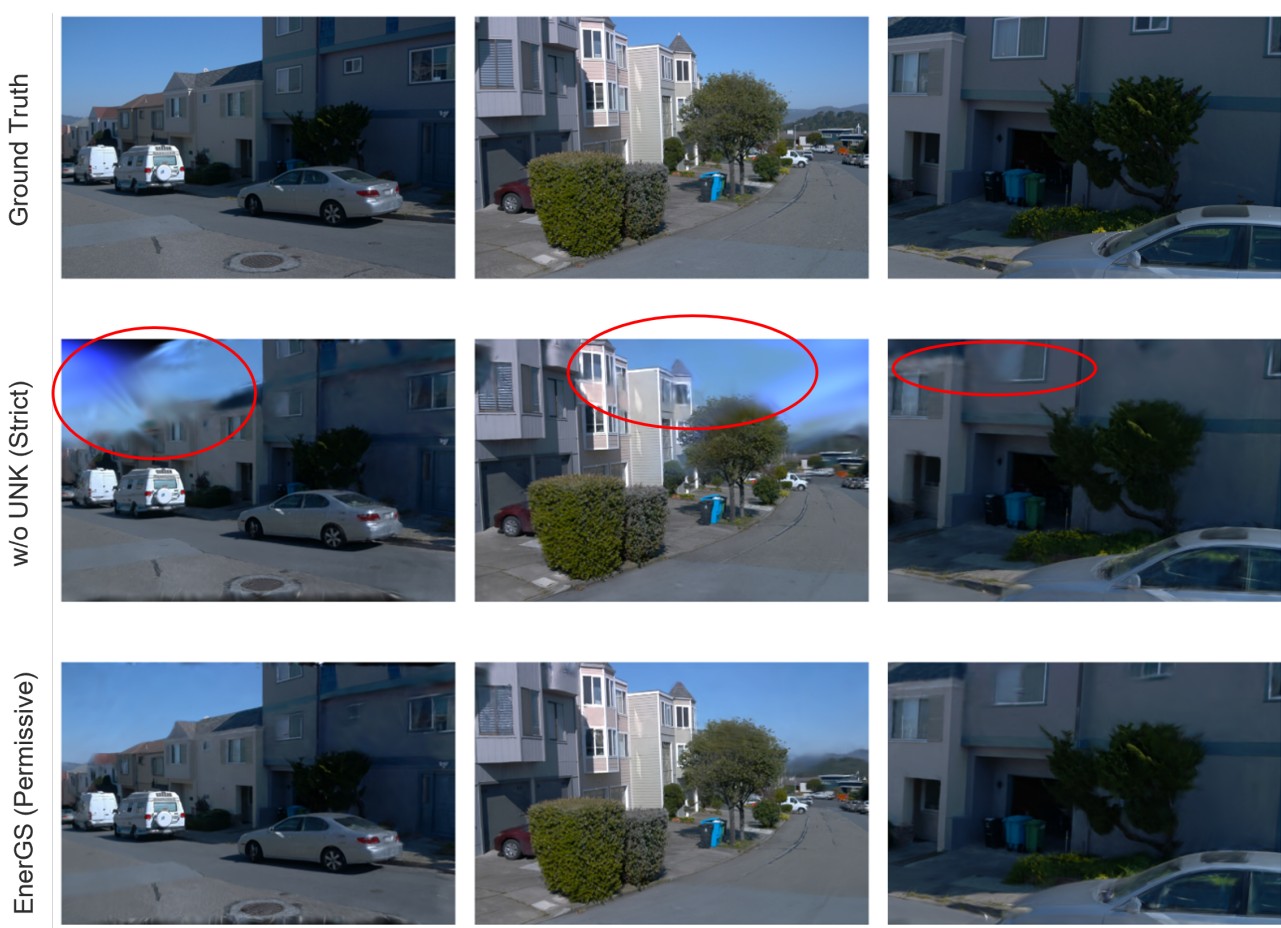

*Figure 6.* Comparison of rendering results in LiDAR blind-spot regions (unobservable geometry), highlighting the effect of enabling the UNK field.

## B. Derivation and Interpretation of Equations (6) and (8)

This section provides the rigorous mathematical derivation and physical interpretation of the energy potentials used in our optimization framework. We specifically detail the gradients for the occupancy energy (attraction force) and the free space

energy (repulsion force).

### B.1. Occupancy Force: Derivation of Equation (6)

To model the surface likelihood robustly against outliers, we employ the Welsch M-estimator. The occupancy energy potential $E_{\text{occ}} : \mathbb{R}^3 \to \mathbb{R}$ is defined as:

$$E_{\text{occ}}(\mu) = -w_{\text{occ}} \exp\left(-\frac{\|\mu - p_{\text{nn}}\|^2}{2\sigma_{\text{occ}}^2}\right), \tag{16}$$

where $p_{\text{nn}}$ denotes the nearest neighbor point in the point cloud $\mathcal{P}$ for the query point $\mu$.

The force acting on $\mu$ is defined as the negative gradient of this potential. Below, we derive the analytical form of this force.

Let $u(\mu)$ be the exponent term:

$$u(\mu) = -\frac{\|\mu - p_{\text{nn}}\|^2}{2\sigma_{\text{occ}}^2}. \tag{17}$$

Thus, the energy can be written as the composite function $E_{\text{occ}} = -w_{\text{occ}} \exp(u)$. Applying the chain rule:

$$\nabla_\mu E_{\text{occ}} = \frac{\partial E_{\text{occ}}}{\partial u} \cdot \nabla_\mu u. \tag{18}$$

Differentiating the exponential function with respect to $u$:

$$\frac{\partial E_{\text{occ}}}{\partial u} = -w_{\text{occ}} \exp(u) = E_{\text{occ}}(\mu). \tag{19}$$

Using the vector calculus identity $\nabla_x \|x - c\|^2 = 2(x - c)$, we differentiate $u$ with respect to $\mu$:

$$\nabla_\mu u = -\frac{1}{2\sigma_{\text{occ}}^2} \nabla_\mu \|\mu - p_{\text{nn}}\|^2 = -\frac{1}{\sigma_{\text{occ}}^2}(\mu - p_{\text{nn}}). \tag{20}$$

Combining these terms, the gradient is:

$$\nabla_\mu E_{\text{occ}} = E_{\text{occ}}(\mu) \cdot \left[-\frac{1}{\sigma_{\text{occ}}^2}(\mu - p_{\text{nn}})\right]. \tag{21}$$

The force is the negative gradient, leading to **Equation (6)**:

$$\mathbf{F}_{\text{occ}}(\mu) = -\nabla_\mu E_{\text{occ}} = \frac{1}{\sigma_{\text{occ}}^2} E_{\text{occ}}(\mu)(\mu - p_{\text{nn}}). \tag{22}$$

Since $E_{\text{occ}}(\mu)$ is always negative, the scalar coefficient $\frac{1}{\sigma^2} E_{\text{occ}}$ is negative. Multiplying this by the vector $(\mu - p_{\text{nn}})$ results in a force vector pointing *towards* the nearest neighbor $p_{\text{nn}}$.

- *Near Field:* When $\mu \approx p_{\text{nn}}$, the exponential term is near 1, providing a strong linear attraction similar to an $L_2$ loss.

- *Far Field:* When $\|\mu - p_{\text{nn}}\| \gg \sigma_{\text{occ}}$, the exponential term decays to zero. This "bounded influence" property ensures that distant outliers exert negligible gradients, preventing the mesh from being distorted by noise.

### B.2. Free Space Barrier: Derivation of Equation (8)

In certified free space $\Omega_{\text{free}}$, the existence of a surface primitive is a physical violation. We model the probability of a "valid" surface existing at penetration depth $s$ using a **Boltzmann distribution** $P(s) \propto \exp(-s/\tau)$. This implies that the associated energy cost must be linear with respect to depth ($E \propto s$).

To ensure $C^1$ continuity for numerical optimization, we implement this linear penalty using a Softplus barrier:

$$E_{\text{free}}(\mu) = \lambda_{\text{free}} \cdot \text{softplus}\left(\frac{d_{\text{trust}}(\mu) - \delta}{\tau}\right), \tag{23}$$

where $\text{softplus}(x) = \ln(1 + e^x)$.

We derive the gradient of $E_{\text{free}}$ with respect to $\mu$. Let the normalized penetration depth be $x(\mu) = \frac{d_{\text{trust}}(\mu) - \delta}{\tau}$.

Applying the chain rule:

$$\frac{\partial E_{\text{free}}}{\partial \mu} = \lambda_{\text{free}} \cdot \frac{\partial \text{softplus}(x)}{\partial x} \cdot \frac{\partial x}{\partial \mu}. \tag{24}$$

Recall that the derivative of the Softplus function is the Sigmoid function $\sigma(\cdot)$:

$$\frac{d}{dx} \ln(1 + e^x) = \frac{e^x}{1 + e^x} = \sigma(x). \tag{25}$$

Differentiating $x(\mu)$ with respect to $\mu$:

$$\frac{\partial x}{\partial \mu} = \frac{1}{\tau} \frac{\partial d_{\text{trust}}}{\partial \mu}. \tag{26}$$

Substituting these back into the chain rule yields **Equation (8)**:

$$\frac{\partial E_{\text{free}}}{\partial \mu} = \frac{\lambda_{\text{free}}}{\tau} \sigma\left(\frac{d_{\text{trust}} - \delta}{\tau}\right) \frac{\partial d_{\text{trust}}}{\partial \mu}. \tag{27}$$

This formulation provides a differentiable "switch" mechanism for outlier rejection:

- **Inside Free Space ($d_{\text{trust}} \gg \delta$):** The Sigmoid term $\sigma(\cdot) \approx 1$. The force becomes a constant repulsive force of magnitude $\lambda/\tau$, pushing the primitive out of the empty space along the normal direction $\frac{\partial d}{\partial \mu}$.

- **Outside Free Space ($d_{\text{trust}} \ll \delta$):** The Sigmoid term $\sigma(\cdot) \approx 0$. The force vanishes smoothly, ensuring that valid surfaces are not penalized.

## C. Further Discussion on Decoupled Optimization vs Joint Optimization

We adopt a decoupled optimization strategy to fundamentally resolve the *gradient conflict* between photometric consistency and physical validity.

In a standard joint optimization framework, the objective is typically formulated as a weighted sum $\mathcal{L}_{\text{total}} = \mathcal{L}_{\text{photo}} + \lambda \mathcal{L}_{\text{geo}}$. However, in sparse-view regimes, this formulation faces two critical issues:

**Domination of Photometric Ambiguity:** As stated in Assumption 1, degenerate solutions (floaters) often yield minimal photometric residuals. Consequently, the gradients derived from $\mathcal{L}_{\text{photo}}$ can create strong energy barriers that oppose the geometric regularization. If the photometric term dominates, the optimizer may become trapped in a local minimum where artifacts are preserved to satisfy image consistency, effectively neutralizing the repulsive force required to clean the map.

**Soft Regularization vs. Hard Projection:** Joint optimization treats the free space constraint as a soft penalty. In contrast, our decoupled approach treats sensor observability as a hard physical constraint. By isolating the geometric update, we ensure that the expulsion force derived in Theorem 1 is applied unconditionally. This acts as a *projection operator* that strictly enforces the trusted domain partition (Assumption 2), guaranteeing that primitives are evicted from certified free space regardless of their contribution to the photometric loss.

Therefore, decoupling is essential to ensure the convergence stability proven in Theorem 1, preventing valid physical constraints from being overridden by ill-posed photometric gradients.

## D. Implementation Details and Hyperparameter Settings

We provide our implementation and summarize the correspondence between the key components described in the main paper and their code implementation.

## D.1. Implementation Overview

The core implementation has two primary modules:

- `geometric_energy.py`: Implements the geometric energy field $E_{\text{geom}}$ and force computation $\mathbf{F} = -\nabla E_{\text{geom}}$.

- `gaussian_model.py`: Extends the base 3DGS model with energy-guided position updates and gradient decoupling.

## D.2. Energy Computation

Listing 1 shows the implementation of energy computation corresponding to Equations (5), (7), and (9) in the main paper.

```python
# E_occ: Welsch M-estimator attraction (Eq. 5)
E_occ = -self.w_occ * torch.exp(-(d_occ**2) / (2 * self.sigma_occ**2))

# E_unk: Weak prior in unknown space (Eq. 9)
E_unk = -self.w_unk * torch.exp(-(d_unk**2) / (2 * self.sigma_unk**2))

# E_free: Softplus barrier in free space (Eq. 7)
E_free = F.softplus((d_trust - self.delta) / self.tau)
E_free = E_free * is_free.float()

# Total geometric energy
E_geom = E_occ + E_unk + self.lambda_free * E_free
```

*Listing 1.* Energy computation for $E_{\text{occ}}$, $E_{\text{unk}}$, and $E_{\text{free}}$.

## D.3. Gradient Computation

Listing 2 shows the gradient computation corresponding to Equations (6) and (8).

```python
# Gradient of E_occ (Eq. 6)
exp_occ = torch.exp(-(d_occ**2) / (2 * self.sigma_occ**2))
coef_occ = (self.w_occ / self.sigma_occ**2) * d_occ * exp_occ
grad_E_occ = coef_occ.unsqueeze(-1) * grad_d_occ

# Gradient of E_unk (same form as E_occ)
exp_unk = torch.exp(-(d_unk**2) / (2 * self.sigma_unk**2))
coef_unk = (self.w_unk / self.sigma_unk**2) * d_unk * exp_unk
grad_E_unk = coef_unk.unsqueeze(-1) * grad_d_unk

# Gradient of E_free: phi'(s) = (1/tau) * sigmoid((s-delta)/tau) (Eq. 8)
phi_prime = (1.0 / self.tau) * torch.sigmoid((d_trust - self.delta) / self.tau)
grad_E_free = phi_prime.unsqueeze(-1) * grad_d_trust * is_free.float().unsqueeze(-1)

# Force: F = -grad(E_geom)
force = -grad_E_occ - grad_E_unk - self.lambda_free * grad_E_free
```

*Listing 2.* Geometric gradient computation for force field.

## D.4. Decoupled Optimization

Listing 3 shows the decoupled position update corresponding to Equation (10), where positions are updated solely by the geometric gradient.

```python
def relax_step(self, ...):
    with torch.no_grad():
        xyz = self.get_xyz

        # Compute force: F = -grad(E_geom)
        force = self.energy_field.compute_force(xyz, coverage_field)
```

```
8          # Position update: mu <- mu + eta * F = mu - eta * grad(E_geom)
9          delta = self.relax_lr * force
10
11         # Step clipping for stability
12         max_step = self.energy_field.voxel_size * 0.5
13         delta = torch.where(delta.norm(dim=-1, keepdim=True) > max_step,
14                             delta * max_step / (delta.norm(dim=-1, keepdim=True) + 1e-8),
15                             delta)
16
17         # Apply update (Eq. 10)
18         self._xyz.data.add_(delta)
```

*Listing 3.* Decoupled position update via geometric energy (Eq. 10).

The photometric gradient is blocked from flowing to positions via the xyz_freeze_in_relax flag, ensuring the decoupling described in Section 3.3:

```
1  # Block photometric gradient on xyz (Section 3.3)
2  self.xyz_freeze_in_relax = True
3
4  def get_xyz_grad_mask_for_relax(self):
5      if self.xyz_freeze_in_relax and self.energs_enabled:
6          return torch.zeros_like(self._xyz)  # Block photometric gradient
7      return torch.ones_like(self._xyz)
```

*Listing 4.* Photometric gradient blocking for decoupled optimization.

### D.5. Discrete Pruning

Listing 5 shows the discrete pruning mechanism corresponding to Equation (12).

```
1  def prune_free_gaussians(self, min_iterations: int = 1000):
2      # Query region for each Gaussian: 1=OCC, 2=FREE, 3=UNK
3      regions = self.energy_field.query_region(self.get_xyz)
4
5      # Prune Gaussians in FREE space (Eq. 12)
6      free_mask = (regions == 2)
7      if free_mask.sum() > 0:
8          self.prune_points(free_mask)
```

*Listing 5.* Discrete pruning of Gaussians in free space (Eq. 12).

### D.6. Method-Code Correspondence Summary

Table 3 summarizes the complete mapping between paper equations and code.

*Table 3.* Correspondence between paper components and code implementation.

| Paper Component | Equation | Implementation |
| --- | --- | --- |
| Occupied attraction $E_{\text{occ}}$ | Eq. (5) | Listing 1, Line 2 |
| Free space barrier $E_{\text{free}}$ | Eq. (7) | Listing 1, Lines 8–9 |
| Unknown regularizer $E_{\text{unk}}$ | Eq. (9) | Listing 1, Line 5 |
| Gradient $\nabla E_{\text{occ}}$ | Eq. (6) | Listing 2, Lines 2–4 |
| Gradient $\nabla E_{\text{free}}$ | Eq. (8) | Listing 2, Lines 12–13 |
| Decoupled position update | Eq. (10) | Listing 3 |
| Gradient decoupling | Sec. 3.3 | Listing 4 |
| Discrete FREE pruning | Eq. (12) | Listing 5 |

### D.7. Default Hyperparameters

*Table 4.* Default hyperparameters for the geometric energy field.

| Energy Term | Parameter | Symbol | Default | Unit |
|---|---|---|---|---|
| $E_{\text{occ}}$ | Weight | $w_{\text{occ}}$ | 1.0 | – |
| | Bandwidth | $\sigma_{\text{occ}}$ | 1.0 | m |
| $E_{\text{unk}}$ | Weight | $w_{\text{unk}}$ | 0.25 | – |
| | Bandwidth | $\sigma_{\text{unk}}$ | 2.0 | m |
| $E_{\text{free}}$ | Barrier weight | $\lambda$ | 1.0 | – |
| | Margin | $\delta$ | 0.5 | m |
| | Temperature | $\tau$ | 0.5 | – |

## E. Sensitivity to Hyperparameters

We evaluate hyperparameter sensitivity across all core components of EnerGS and found the method to be broadly stable: most perturbations cause only negligible changes, while the only clear degradation arises in extreme settings that explicitly violate the intended OCC/UNK design principle.

**Clarification of Parameter Coupling.** The attractive energy gradients share the same form. In the near-surface regime ($d \to 0$), the force becomes $\mathbf{F} \approx -\frac{w}{\sigma^2}d$, where $\frac{w}{\sigma^2}$ is the effective spring constant. We vary $\sigma$ by $2\times$ while fixing $\frac{w}{\sigma^2}$, and observe only small changes on both datasets: OCC gives an averaged PSNR std/range of 0.04 / 0.10 dB, and UNK 0.05 / 0.12 dB. This supports coupling $w$ with $\sigma^2$: since most Gaussians converge near target surfaces ($d \to 0$), the effective stiffness $\frac{w}{\sigma^2}$, not the far-field potential shape, dominates. The 7 nominal hyperparameters thus reduce to 5 independent ones.

**Sensitivity to Parameter Perturbation.** We sweep parameters individually (all others fixed). Except for the extreme case where $w_{\text{unk}}$ increases from 0.0625 to 1.0 (a $16\times$ increase in the UNK effective force), all parameters show PSNR Std below 0.07 dB, well within the stochastic training noise floor. On Waymo and KITTI, which differ fundamentally in LiDAR density, sensor configuration, and urban layout, their sensitivity profiles are broadly consistent.

**Robustness of Discrete Pruning Threshold.** The discrete pruning threshold $\tau_{\text{margin}}$ is not highly sensitive. On both Waymo and KITTI, performance remains stable across $\tau_{\text{margin}} \in [0.1, 2.0]$. The primary free-space constraint comes from the continuous energy field, while discrete pruning acts only as a secondary cleanup step.

**Extreme OCC/UNK Ratios.** We test the relative strength between OCC and UNK attraction by fixing the geometric mean and sweeping the ratio $R = \frac{w_{\text{occ}}/\sigma_{\text{occ}}^2}{w_{\text{unk}}/\sigma_{\text{unk}}^2}$ from UNK- to OCC-dominant. Violating the physical prior ($R < 1$) degrades PSNR consistently on both datasets, confirming that OCC surfaces should attract more strongly than UNK regions.

*Table 5.* **Extreme OCC/UNK ratios for the validation of the design principle.**

| R | Interpretation | Waymo $\Delta$PSNR | KITTI $\Delta$PSNR |
|---|---|---|---|
| 1/16 | UNK $\gg$ OCC | -0.66 | -0.23 |
| 1/4 | UNK $>$ OCC | -0.29 | -0.10 |
| 4 | OCC $>$ UNK | -0.05 | -0.07 |
| 16 | OCC $\gg$ UNK | 0.00 | 0.00 |
| 64 | OCC dominant | -0.05 | -0.01 |

## F. Additional LiDAR-Prior-Based Baseline

We integrate our energy-based regularization into the gsplat-based pipeline of SplatAD [16], a recent LiDAR-guided 3DGS method. As shown in Tab. 6, EnerGS consistently improves both photometric quality and geometric regularity when combined with SplatAD. On KITTI, where LiDAR points are sparser, EnerGS achieves significant gains, notably reducing the Leak score from 22.1 to 7.0. On Waymo, where the LiDAR prior is denser, the performance remains comparable while

still reducing leakage artifacts. This confirms that EnerGS is particularly beneficial under sparse geometric priors while remaining compatible with stronger LiDAR-based pipelines.

*Table 6.* **Comparison with SplatAD.** EnerGS improves performance when integrated into a LiDAR-prior-based pipeline, particularly on sparser datasets like KITTI.

| Dataset | Method | PSNR↑ | SSIM↑ | OccCov↑ | Margin↑ | Thick↓ | Leak↓ |
|---------|--------|-------|-------|---------|---------|--------|-------|
| KITTI | SplatAD | 17.14 | 0.521 | **59.2** | 1.55 | 1.38 | 22.1 |
| | EnerGS+SplatAD | **18.32** | **0.545** | 37.6 | **1.68** | 1.44 | **7.0** |
| Waymo | SplatAD | 26.20 | **0.787** | 44.3 | **3.27** | **0.73** | 1.4 |
| | EnerGS+SplatAD | **26.34** | 0.784 | **44.5** | 3.22 | **0.73** | **1.2** |

## G. Computational Efficiency

We measure the rendering and training speed on an NVIDIA L40S GPU (Tab. 7). EnerGS maintains real-time rendering performance (102.2 FPS on KITTI, 72.9 FPS on Waymo) and introduces only limited training overhead compared to vanilla 3DGS. The geometric prior initialization is a one-time preprocessing step taking ∼1.6s on KITTI and ∼5.5s on Waymo. This confirms that EnerGS provides improved geometric fidelity with a reasonable computational trade-off.

*Table 7.* **Efficiency Comparison.** Rendering speed (FPS) and training speed (wall step/s) on an NVIDIA L40S GPU.

| Method | Rendering (FPS)↑ | | Training (step/s)↑ | |
|--------|-------|-------|-------|-------|
| | KITTI | Waymo | KITTI | Waymo |
| 3DGS | 133.5 | 85.3 | 42.39 | 19.29 |
| 2DGS | 86.5 | 49.8 | 34.40 | 15.89 |
| Taming-3DGS | 193.6 | 114.5 | 106.71 | 63.51 |
| GeoGaussian | 221.5 | 70.5 | 62.49 | 24.17 |
| Mip-Splatting | 107.2 | 58.8 | 52.42 | 23.14 |
| Scaffold-GS | 61.7 | 86.2 | 17.51 | 11.77 |
| EnerGS (Ours) | 102.2 | 72.9 | 39.44 | 17.96 |

## H. Compatibility with Dynamic Scene Modeling

While EnerGS focuses on static 3D reconstruction, it is complementary to dynamic Gaussian Splatting pipelines for autonomous driving. Recent methods handle dynamic scenes through different strategies, including foreground-background decomposition in [51, 43], deformation-based dynamic modeling in [21, 34], and object-aware representations in [42]. To show the compatibility with dynamic scene modeling, we integrate EnerGS with the dynamic Gaussian framework StreetGS [43]. As shown in Tab. 8, EnerGS improves the static-region reconstruction (PSNR from 25.86 to 26.22) while maintaining comparable performance in dynamic regions, demonstrating its plug-and-play advantage.

*Table 8.* **Integration with Dynamic Scene Modeling.** Comparison with StreetGS on static and dynamic regions.

| Method | PSNR (Full)↑ | SSIM (Full)↑ | PSNR (Dyn)↑ | SSIM (Dyn)↑ | PSNR (Stat)↑ | SSIM (Stat)↑ |
|--------|--------------|--------------|-------------|-------------|--------------|--------------|
| StreetGS | 25.58 | 0.819 | **25.11** | **0.764** | 25.86 | 0.826 |
| EnerGS (Ours) | **25.94** | **0.826** | 25.10 | 0.762 | **26.22** | **0.834** |

