# OpenReview forum: "EnerGS: Energy-Based Gaussian Splatting under Partial Geometric Priors"
_ICML.cc/2026/Conference — ICML 2026 regular_

### Official Review · Reviewer_tpuz · 2026-02-26

**Soundness:** 3
**Presentation:** 3
**Significance:** 2
**Originality:** 3
**Overall Recommendation:** 4
**Confidence:** 4

**Summary:**

This paper introduces EnerGS, a framework designed to enhance 3DGS for large-scale outdoor scenes by incorporating partial geometric priors. Instead of treating geometric data as hard constraints, EnerGS models the scene as a continuous probabilistic energy field that adaptively attracts primitives to occupied regions, repels them from free space, and applies weak priors in unobserved areas. The authors propose a decoupled optimization strategy that updates Gaussian positions solely via geometric gradients while optimizing appearance via photometric loss. Experiments on KITTI and Waymo datasets demonstrate that EnerGS achieves superior reconstruction quality.

**Compliance With Llm Reviewing Policy:**

Affirmed.

**Final Justification:**

The rebuttal has addressed my main concerns. My final recommendation is weak accept (4).

**Key Questions For Authors:**

1. This paper explicitly block the photometric gradient from updating the position μ. While this prevents floaters, does it also prevent the model from achieving sub-pixel alignment in high-texture areas where the camera signal is more precise than the discretized LiDAR voxel grid?
2. The current baselines (2DGS, Scaffold-GS) are general 3DGS variants. Is it possible to include some specialized LiDAR-guided models (like LiGSM[1] or DrivingGaussian[2]) in the quantitative comparison for KITTI/Waymo?
3. Could the authors provide a quantitative analysis of the computational overhead for the proposed approach?

Ref:
[1] Shen, Jian, et al. "LiDAR-enhanced 3D Gaussian Splatting Mapping." 2025 IEEE International Conference on Robotics and Automation (ICRA). IEEE, 2025.
[2] Zhou, Xiaoyu, et al. "Drivinggaussian: Composite gaussian splatting for surrounding dynamic autonomous driving scenes." Proceedings of the IEEE/CVF conference on computer vision and pattern recognition. 2024.

**Limitations:**

The author should provide some discussions on the potential limitations of the paper (such as sensitivity to LiDAR quality, etc.).

**Strengths And Weaknesses:**

Strengths:
1. The paper formulates LiDAR priors as a continuous, differentiable energy field using Welsch M-estimators (attraction) and Boltzmann barriers (repulsion), which is novel.
2. The formulation of the problem and theoretical analysis is quite sound.
3. The experimental results and ablations convincingly proved the effectiveness of the approach.

Weaknesses:
1. The evaluation explicitly excludes dynamic objects, focusing solely on static scene reconstruction. This simplification limits the framework's immediate applicability to real-world, unfiltered autonomous driving scenarios where dynamic objects are inevitable.
2. Because photometric gradients are blocked from updating positions, the method relies heavily on accurate LiDAR calibration; sensor noise or misalignment could potentially lead to texture artifacts that the photometric loss cannot correct.
3. The heavy reliance on geometric regularization in occupied regions might suppress high-frequency geometric details that are visible in images but not captured by the sparse LiDAR point cloud.

---

> ### Author Rebuttal · Authors · 2026-03-31
>
> Thank you for the constructive review. We use **(W)** for points from **Weaknesses** and **(Q)** for **Key Questions**.
>
> **1. Discussion and extensions on dynamic objects (W1)**
>
> We would like to clarify that 3D reconstruction and 4D dynamic scene modeling are two related but distinct problems. EnerGS is designed to address the former. While 4D is highly important for autonomous driving, 3D reconstruction itself remains a well-motivated and practically important problem: 1) many dynamic-scene methods still rely on static–dynamic decomposition, and 2) in photorealistic closed-loop simulation, one often prioritizes high-quality static 3D scene reconstruction while handling dynamic agents separately as controllable actors.
>
> Many recent autonomous-driving Gaussian methods adopt a foreground–background decomposition strategy, such as DrivingGaussian (CVPR24), StreetGS (ECCV24), DeSiRe-GS (CVPR25), and AD-GS (ICCV25). EnerGS is complementary to this line of work and can be naturally integrated into such pipelines to improve the static/background reconstruction component.
>
> We also demonstrate this compatibility and provide a visualization of the static–dynamic decomposition in the [`video`](https://anonymous.4open.science/r/EnerGS_ICML26_Rebuttal-97AF/EnerGS_Dynamic.mp4), where EnerGS is combined with a dynamic Gaussian framework. We also compare EnerGS with StreetGS.
>
> | Experiment | PSNR (Full) ↑ | SSIM (Full) ↑ | PSNR (Dynamic) ↑ | SSIM (Dynamic) ↑ | PSNR (Static) ↑ | SSIM (Static) ↑ |
> |---|---:|---:|---:|---:|---:|---:|
> | StreetGS | 25.58 | 0.819 | 25.11 | 0.764 | 25.86 | 0.826 |
> | EnerGS | 25.94 | 0.826 | 25.10 | 0.762 | 26.22 | 0.834 |
>
> The results show that while the dynamic-region performance is comparable, EnerGS provides a clear improvement in static-region reconstruction, which is consistent with our design focus.
>
> **2. Effect of geometric regularization on high-frequency details (W2, W3, Q2)**
>
> Blocking the direct photometric gradient on $\mu$ does not necessarily harm high-frequency reconstruction, especially when the LiDAR geometry is already structurally reasonable. In 3DGS, high-frequency details are not determined solely by the sub-pixel optimization of Gaussian centers, but are largely represented through Gaussian scale/variance, opacity, appearance parameters, and the superposition of multiple Gaussians. Therefore, moderate discretization error from the LiDAR voxel grid does not directly lead to distortions in high-texture regions. Moreover, photometric signals still influence the spatial representation through densification, which can introduce new Gaussians in regions with high residuals. Under a reasonable LiDAR prior, the proposed energy field further improves structural stability by discouraging physically implausible floaters, while high-frequency rendering details remain mainly governed by appearance and density adaptation.
>
> We further analyze high- and low-texture regions using a Laplacian-based score and find that EnerGS improves PSNR by +0.55 dB in high-texture regions and +0.90 dB in low-texture regions, confirming that our design does not degrade high-frequency details.
>
> **3. Comparison with LiDAR-guided models (Q2)**
>
> Neither LiGSM (ICRA 2025) nor DrivingGaussian (CVPR 2024) has directly released its code. We therefore compare with the more recent LiDAR-guided baseline SplatAD (CVPR 2025), as suggested by Reviewer vC8T. EnerGS improves over SplatAD on KITTI in PSNR, SSIM, Margin, and Leak, while remaining comparable overall on Waymo. This is consistent with our goal of improving robustness under partial geometric priors, especially in sparser LiDAR settings. Please refer to our response to (W1) from Reviewer vC8T.
>
> **4. Quantitative analysis of the computational cost (Q3)**
>
> We have included a quantitative analysis of the training overhead on an NVIDIA L40S GPU. A rendering-speed comparison can be found in the answer to (W1, Q1) from Reviewer 8wdk.
>
> | Method | KITTI (wall step/s) ↑ | Waymo (wall step/s) ↑ |
> |---|---:|---:|
> | 3DGS | 42.39 | 19.29 |
> | 2DGS | 34.40 | 15.89 |
> | Taming-3DGS | 106.71 | 63.51 |
> | GeoGaussian | 62.49 | 24.17 |
> | Mip-Splatting | 52.42 | 23.14 |
> | Scaffold-GS | 17.51 | 11.77 |
> | EnerGS (ours) | 39.44 | 17.96 |
>
> EnerGS introduces only limited training overhead compared with vanilla 3DGS, while remaining competitive with several representative baselines. Specifically, the training speed decreases from 42.39 to 39.44 wall step/s on KITTI and from 19.29 to 17.96 wall step/s on Waymo.
>
> **Initialization cost**: The geometric prior initialization is a one-time preprocessing step implemented with Numba, taking 1.607 ± 0.806 s on KITTI and 5.519 ± 0.905 s on Waymo on the same GPU. Note that this initialization cost is required only during training.
>
> Overall, these results indicate that EnerGS adds only limited extra computation while providing improved geometric fidelity and better free-space regularization. We will include this analysis in the revised paper.

---

> > ### Author Rebuttal · Reviewer_tpuz · 2026-04-03
> >
> > I appreciate the detailed reply. The additional experiments on LiDAR-based approach and computational cost have improved the quality of the paper. I will thus raise my score to 4.

---

> > > ### Author Response · Authors · 2026-04-03
> > >
> > > Thank you very much for your positive feedback and for raising your score for our paper. We are very glad to know that your concerns have been fully resolved. We will incorporate these additional experiments into the final version to further improve the quality of the paper. We sincerely appreciate all your suggestions.
> > >
> > > We would also be happy to participate in any further discussion if needed.

---

### Official Review · Reviewer_2XBB · 2026-03-11

**Soundness:** 3
**Presentation:** 2
**Significance:** 2
**Originality:** 3
**Overall Recommendation:** 4
**Confidence:** 2

**Summary:**

This paper proposes EnerGS, an energy-driven 3D Gaussian Splatting (3DGS) framework designed for large-scale outdoor scenes. To address the optimization challenges caused by sparse and incomplete geometric priors (such as LiDAR data), the authors explicitly divide 3D space into Occupied, Free, and Unknown regions, constructing a continuous geometric energy field based on this division. Furthermore, the paper introduces a decoupled optimization strategy that separates the geometric position updates of Gaussian primitives from photometric appearance optimization. This approach allows unknown regions to be reconstructed using photometric information while strictly eliminating geometric artifacts in free space. Extensive experiments on the KITTI and Waymo datasets demonstrate that this method achieves state-of-the-art performance in both photometric rendering quality and geometric structural stability.

**Compliance With Llm Reviewing Policy:**

Affirmed.

**Final Justification:**

I appreciate the authors’ detailed response and the additional analyses provided in the rebuttal. The clarifications help improve the clarity of the work and address several of my questions.

**Key Questions For Authors:**

- During initialization, the method uses a voxel grid with a resolution of $V^3$ to pre-compute distance transformations. For city-scale environments, will this fixed-resolution grid encounter VRAM or computational bottlenecks? Are there any experimental tests regarding the impact of different grid resolutions on final reconstruction accuracy?
- Regarding the Discrete Pruning strategy, it relies on a hard boundary threshold $\tau_{margin}$ to clear Gaussian spheres in free space. Does this threshold require fine-tuning when applied to other LiDAR datasets with different sparsity levels and noise distribution characteristics?
- The paper explicitly notes that current evaluations exclude dynamic objects. How can (or can) the EnerGS framework be extended to scenes with dynamic objects in the future? Would this violate the "Free Space" assumption in the current static energy field?

**Limitations:**

Yes.

**Strengths And Weaknesses:**

- Strengths
   - Strong Originality: The introduction of a probabilistic energy field to handle incomplete geometric priors is highly innovative. The method creatively applies the Welsch M-estimator for occupied space, a Boltzmann barrier for free space, and treats unknown regions as weak priors.
   - Soundness and Rigor: The authors propose an effective decoupled optimization rule to resolve conflicts between geometric and photometric gradients in occluded areas. The paper provides rigorous theoretical proofs (e.g., Theorem 1 proves the instability of degenerate solutions in free space; Theorem 2 proves the regularity of optimization trajectories), demonstrating a solid mathematical foundation.
   - Significance and Practicality: The method directly addresses the critical pain point of geometric collapse caused by sparse viewpoints in large-scale, unbounded outdoor scenes (e.g., autonomous driving). Benchmark results (KITTI, Waymo) show a significant reduction in geometric leakage, indicating high application potential.
   - Clear Presentation: The paper features tight logic and excellent illustrations. The supplementary material provides clear and detailed derivations of the energy functions, making the core ideas easy to understand and reproduce.

- Weaknesses:
  - The current evaluation completely excludes dynamic objects and focuses solely on static scenes, which is a notable limitation for real-world autonomous driving applications.
  - The construction of geometric priors relies on a pre-computed discrete voxel grid with a resolution of $V^3$ to perform Euclidean distance transformations. In extremely large-scale scenes, this global meshing might introduce significant memory overhead, a potential issue that lacks detailed discussion in the tex- The construction of geometric priors relies on a pre-computed discrete voxel grid with a resolution of $V^3$ to perform Euclidean distance transformations.

---

> ### Author Rebuttal · Authors · 2026-03-31
>
> Thank you for your thoughtful review and for raising these important questions. We use **(W)** for points from **Weaknesses** and **(Q)** for **Key Questions**.
>
> **1. City-scale environments, memory overhead, and computational costs (W2, Q1)**
>
> We agree that using a single dense global voxel grid may become memory-intensive for city-scale scenes. However, city-scale Gaussian-based 3D reconstruction is usually not handled with a single voxel grid. We would like to point out that city-scale Gaussian reconstruction is generally considered a different problem setting, whose main focus is on how to scale to very large environments through techniques such as scene partitioning, chunk-wise processing, or hierarchical decomposition, where the distance field is computed locally rather than globally. Existing works such as DoGaussian (NeurIPS 2024), CityGaussian (ECCV 2024), and Horizon-GS (CVPR 2025) mainly focus on efficient distributed processing and integration of large numbers of Gaussians, and some of them are also designed more for aerial-to-ground setups. In contrast, outdoor or autonomous-driving Gaussian splatting methods are usually concerned with reconstructing scenes more accurately from trajectory-based camera inputs. The typical scale in this setting is scene-level, often under 100 frames. For sequences longer than 100 frames, many existing methods also recommend using multiple subsets of Gaussian models rather than a single monolithic model, such as S3Gaussian (arXiv 2024) and CoDa-4DGS (ICCV 2025).
>
> **2. Hard boundary threshold and different LiDAR sparsity (Q2)**
>
> We note that Discrete Pruning is not the primary free-space constraint in our method. The main effect comes from the energy field, which already pushes Gaussians out of invalid free-space regions during optimization (Theorem 1 and Appendix A.1). Discrete Pruning only acts as a secondary cleanup step to remove erroneous Gaussians introduced by densification, especially in unstable early training.
>
> Accordingly, $\tau_{margin}$ is not a highly sensitive hyperparameter and does not require delicate dataset-specific tuning in our experiments. On both Waymo and KITTI (2 scenes each), despite their substantially different LiDAR sparsity, performance remains stable across $\tau_{margin} \in [0.1, 2.0]$: PSNR varies only from 26.45 to 26.57 on Waymo and from 16.44 to 16.58 on KITTI, while Leak% stays within 1.3–1.5 and 5.3–6.0, respectively.
>
> **Waymo**
>
> | $\tau_{margin}$ | PSNR ↑ | OccCov ↑ | Margin ↑ | Leak ↓ | #G |
> |---|---:|---:|---:|---:|---:|
> | 0.1 | 26.45 | 37.3% | 5.133 | 1.3% | 1.92M |
> | 0.3 | 26.54 | 38.2% | 5.224 | 1.3% | 1.91M |
> | 0.5 | 26.57 | 37.7% | 5.092 | 1.5% | 1.94M |
> | 1.0 | 26.51 | 37.8% | 5.151 | 1.3% | 1.93M |
> | 2.0 | 26.49 | 37.9% | 5.092 | 1.5% | 1.95M |
>
> **KITTI**
>
> |$\tau_{margin}$ | PSNR ↑ | OccCov ↑ | Margin ↑ | Leak ↓ | #G |
> |---|---:|---:|---:|---:|---:|
> | 0.1 | 16.52 | 33.3% | 2.344 | 5.8% | 1.26M |
> | 0.3 | 16.52 | 33.7% | 2.863 | 6.0% | 1.29M |
> |0.5 | 16.58 | 33.5% | 3.030 | 5.8% | 1.25M |
> |1.0 | 16.54 | 34.0% | 2.883 | 5.3% | 1.25M |
> |2.0 | 16.44 | 33.8% | 3.209 | 5.6% | 1.29M |
>
> **3. Extensions on dynamic objects compatibility (W1, Q3)**
>
> We would like to clarify that 3D reconstruction and 4D dynamic scene modeling are two related but distinct problems. EnerGS is designed to address the former. While 4D is highly important for autonomous driving, 3D reconstruction itself remains a well-motivated and practically important problem: 1) many dynamic-scene methods still rely on static–dynamic decomposition, and 2) in photorealistic closed-loop simulation, one often prioritizes high-quality static 3D scene reconstruction while handling dynamic agents separately as controllable actors.
>
> Many recent autonomous-driving Gaussian methods adopt a foreground–background decomposition strategy, such as DrivingGaussian (CVPR24), StreetGS (ECCV24), DeSiRe-GS (CVPR25), and AD-GS (ICCV25). EnerGS is complementary to this line of work and can be naturally integrated into such pipelines to improve the static/background reconstruction component.
>
> We also demonstrate this compatibility and provide a visualization of the static–dynamic decomposition in the [`video`](https://anonymous.4open.science/r/EnerGS_ICML26_Rebuttal-97AF/EnerGS_Dynamic.mp4), where EnerGS is combined with a dynamic Gaussian framework. We also compare EnerGS with StreetGS.
>
> | Experiment | PSNR (Full) ↑ | SSIM (Full) ↑ | PSNR (Dynamic) ↑ | SSIM (Dynamic) ↑ | PSNR (Static) ↑ | SSIM (Static) ↑ |
> |---|---:|---:|---:|---:|---:|---:|
> | StreetGS | 25.58 | 0.819 | 25.11 | 0.764 | 25.86 | 0.826 |
> | EnerGS | 25.94 | 0.826 | 25.10 | 0.762 | 26.22 | 0.834 |
>
> The results show that while the dynamic-region performance is comparable, EnerGS provides a clear improvement in static-region reconstruction, which is consistent with our design focus.

---

> > ### Author Rebuttal · Reviewer_2XBB · 2026-04-03
> >
> > I appreciate the authors’ detailed response and the additional analyses provided in the rebuttal. The clarifications help improve the clarity of the work and address several of my questions.
> > Overall, my assessment of the paper change to 4.

---

> > > ### Author Response · Authors · 2026-04-03
> > >
> > > Thank you very much for your positive feedback and for updating your assessment of our paper. We are very glad to know that your concerns have been fully resolved.
> > >
> > > We would also be happy to actively participate in any further discussion if needed.

---

### Official Review · Reviewer_vC8T · 2026-03-12

**Soundness:** 3
**Presentation:** 3
**Significance:** 3
**Originality:** 3
**Overall Recommendation:** 4
**Confidence:** 3

**Summary:**

This paper introduces EnerGS, a novel framework that enhances 3D Gaussian Splatting (3DGS) for large-scale outdoor scene reconstruction by effectively integrating incomplete geometric priors, such as sparse LiDAR data. To overcome the issue of "floaters" and overfitting caused by conflicting photometric and geometric gradients in unobserved regions , EnerGS models the scene as a continuous probabilistic geometric energy field. This field logically divides space into occupied regions with robust attraction forces, free space with strict repulsion barriers, and unknown space with permissive weak priors. Crucially, the authors propose a decoupled optimization strategy where Gaussian positions are updated exclusively by the geometric energy gradient to ensure physical validity, while their appearance and shape parameters are driven by photometric loss. Extensive experiments on the KITTI and Waymo datasets demonstrate that EnerGS successfully eliminates free-space artifacts and accurately reconstructs geometry in LiDAR blind spots, achieving state-of-the-art photometric rendering and geometric stability

**Compliance With Llm Reviewing Policy:**

Affirmed.

**Final Justification:**

In summary, while the performance gains demonstrated on the Waymo dataset are undeniably marginal, the proposed methodology is highly novel and introduces a valuable new perspective to the field. I believe the technical innovation outweighs the limited empirical improvements. Therefore, I maintain my stance to lean towards acceptance (Weak Accept / Borderline Accept) to encourage such methodological exploration. However, given the weak quantitative results, I would not object if the Area Chair ultimately decides to reject the paper based on the consensus.

**Key Questions For Authors:**

Show in the Weaknesses

**Limitations:**

Yes

**Strengths And Weaknesses:**

### Strengths

* **High Quality of Writing:** The paper is exceptionally well-written, logically structured, and easy to follow. The theoretical analysis is mathematically sound and clearly supports the proposed methodology, making the transition from the initial motivation to the final implementation highly compelling.
* **Innovative Methodology:** The proposed EnerGS framework introduces a highly innovative approach to geometric regularization. By modeling partial observability as a probabilistic geometric energy field and employing a decoupled optimization strategy, the method elegantly resolves the gradient conflicts that typically plague standard joint optimization techniques in sparse-view settings.

### Weaknesses

* **Missing Relevant Baselines:** The experimental evaluation lacks comparisons against several recent and highly relevant LiDAR-based 3DGS baselines. Although referenced in the related work, the paper would benefit significantly from a direct quantitative and qualitative comparison with methods such as *SplatAD: Real-Time Lidar and Camera Rendering with 3D Gaussian Splatting for Autonomous Driving*. Including these specific autonomous driving-focused baselines would provide a much more rigorous assessment of the proposed method's relative performance and state-of-the-art claims.

---

> ### Author Rebuttal · Authors · 2026-03-31
>
> Thank you for your positive assessment of our paper and for the clear suggestions. We appreciate that your comments are both reasonable and actionable. To keep the rebuttal format aligned and easy to follow for all reviewers, we use **(W)** to denote points raised in **Weaknesses** and **(Q)** to denote items from **Key Questions for Authors**.
>
> **1. Comparison with SplatAD (W1)**
>
> As suggested, we added a comparison with a LiDAR-prior-based method by integrating EnerGS into the SplatAD gsplat-based pipeline:
>
> | Group | Method | PSNR ↑ | SSIM ↑ | OccCov ↑ | Margin ↑ | Thick ↓ | Leak ↓ |
> |---|---|---:|---:|---:|---:|---:|---:|
> | KITTI | SplatAD | 17.14 | 0.521 | 59.2 | 1.55 | 1.38 | 22.1 |
> | KITTI | EnerGS+SplatAD | 18.32 | 0.545 | 37.6 | 1.68 | 1.44 | 7.0 |
> | Waymo | SplatAD | 26.20 | 0.787 | 44.3 | 3.27 | 0.73 | 1.4 |
> | Waymo | EnerGS+SplatAD | 26.34 | 0.784 | 44.5 | 3.22 | 0.73 | 1.2 |
>
> These results show that EnerGS remains effective when combined with a LiDAR-prior-based pipeline. On KITTI, where the LiDAR points are sparser, it consistently improves both photometric quality and geometric regularity, with a notably lower Leak score. On Waymo, the gains are smaller, but the overall performance remains comparable while still reducing leakage artifacts. This is consistent with our motivation: EnerGS is particularly beneficial when the geometric prior is partial or sparse, while remaining compatible with stronger LiDAR-based priors. We will include this comparison in the revised paper.

---

> > ### Author Rebuttal · Reviewer_vC8T · 2026-04-03
> >
> > Thank you for the detailed rebuttal and the additional experiments.
> >
> > I appreciate the effort the authors put into addressing my concerns, particularly in providing the new baseline comparisons with the LiDAR-prior-based method (SplatAD).
> >
> > I have carefully reviewed the newly added results. While the improvements on the KITTI dataset are evident and demonstrate the method's effectiveness under sparse priors, I noticed that the performance gains on the Waymo dataset are quite marginal (e.g., PSNR increasing only from 26.20 to 26.34, with minor improvements in the Leak score). I read your explanation regarding Waymo's denser LiDAR points and the method's primary focus on sparse geometric priors, but the limited performance ceiling on higher-quality datasets remains a point of consideration.
> >
> > I will carefully review the comments from the other reviewers and actively participate in the upcoming reviewer discussion phase before making my final decision on the score.
> >
> > Thanks again for your hard work during the rebuttal period.

---

> > > ### Author Response · Authors · 2026-04-03
> > >
> > > Thank you very much for your thoughtful follow-up comments. We are glad that your concerns have been adequately addressed.
> > >
> > > Regarding the gains on Waymo, we believe this is closely related to the characteristics of SplatAD itself. As a LiDAR-guided method, SplatAD relies heavily on strong geometric support when the LiDAR prior is dense, which naturally leaves less room for further improvement. At the same time, this also suggests a limitation of such methods: their robustness can be more restricted when the geometric prior becomes sparser or less complete. In contrast, EnerGS can better compensate for this limitation through its energy-based optimization, leading to stronger robustness when geometric support is weaker. In addition, EnerGS shows consistent improvements in LiDAR-uncovered regions on both KITTI and Waymo, especially in scenes with tall buildings or trees, where incomplete geometric observations are more common.
> > >
> > > Furthermore, our comparison with SplatAD more clearly highlighted the plug-and-play advantage of EnerGS. Beyond our original implementation based on INRIA-3DGS, which is included in the supplementary material, we also implemented EnerGS on top of the gsplat framework used by SplatAD, as well as on a dynamic Gaussian Splatting framework in response to comments from other reviewers, demonstrating its ease of integration across different Gaussian Splatting codebases.
> > >
> > > Thank you again for your constructive feedback, and helping us better recognize the plug-and-play advantage of our method during the rebuttal process.

---

### Official Review · Reviewer_8wdk · 2026-03-14

**Soundness:** 2
**Presentation:** 3
**Significance:** 3
**Originality:** 3
**Overall Recommendation:** 3
**Confidence:** 3

**Summary:**

This paper introduces EnerGS, a method that helps 3D Gaussian Splatting work better when geometric data is incomplete or unreliable. Instead of forcing strict geometric rules, EnerGS provides soft guidance whic steers optimisation, improving both visual quality and geometric stability. Authors targered their work to large outdoor scenes.

**Compliance With Llm Reviewing Policy:**

Affirmed.

**Key Questions For Authors:**

1. What is the rendering speed?
2. How well EnerGS perform in other viewpoints?
3. What is the performance of proposed method compared with lidar prior based methods?

**Limitations:**

Yes

**Strengths And Weaknesses:**

EnerGS seems technically solid supported by ablations studies. The approach is logically consistent and tested in realistic settings. The paper is well written using illustrative figures and explanations. In overall, the structure helps reader to follow the main ideas of the paper.

EnerGS addresses an important problem in a 3D scene reconstruction under sparse geometry in large outdoor environments. Authors models partial geometry as an energy field and uses it for soft guidance providing originality to work. The improvements suggest meaningful practical impact. However, the paper does not provide information of rendering speed. Also, qualitative results are shown from a few selected viewpoints in which EnerGS provides better results. More viewpoints would be needed for more comprehensive comparison. Additionally, it would be good to have a comparison to lidar-based methods.

---

> ### Author Rebuttal · Authors · 2026-03-31
>
> Thank you for the insightful suggestions and questions. Below, we address the points in **Weaknesses** with **(W)** and the **Key Questions For Authors** with **(Q)**.
>
> **1. Rendering Speed (W1, Q1)**
>
> We measured rendering speed on an NVIDIA L40S GPU:
> | Method | KITTI (FPS) | Waymo (FPS) |
> |---|---:|---:|
> | 3DGS | 133.5 | 85.3 |
> | 2DGS | 86.5 | 49.8 |
> | Taming-3DGS | 193.6 | 114.5 |
> | GeoGaussian | 221.5 | 70.5 |
> | Mip-Splatting | 107.2 | 58.8 |
> | Scaffold-GS | 61.7 | 86.2 |
> | EnerGS (ours) | 102.2 | 72.9 |
>
> EnerGS maintains real-time rendering performance, achieving 102.2 FPS on KITTI and 72.9 FPS on Waymo. While it is slightly slower than vanilla 3DGS, it remains competitive with several GS-based baselines. We believe this is a reasonable trade-off for improved geometric fidelity in sparse outdoor scenes.
>
> **2. More viewpoints (Q2)**
>
> We follow the common evaluation protocol in novel view synthesis and use one test view every four frames for quantitative comparison. To provide a more comprehensive qualitative evaluation, we additionally compare EnerGS under more challenging viewpoint changes, including varying yaw angles ([`Fig.1`](https://anonymous.4open.science/r/EnerGS_ICML26_Rebuttal-97AF/Rotated%20Viewpoints.png)) and lateral offsets ([`Fig.2`](https://anonymous.4open.science/r/EnerGS_ICML26_Rebuttal-97AF/Shifted%20Viewpoints.png)). These results show that EnerGS can consistently improve rendering quality under more diverse novel-view settings, beyond the selected viewpoints shown in the main paper.
>
> **3. LiDAR-prior-based methods(Q3)**
>
> We added a comparison with the LiDAR-based method SplatAD by applying our energy-based regularization on top of its gsplat-based pipeline. The results are shown below:
>
> | Group | Method | PSNR ↑ | SSIM ↑ | OccCov ↑ | Margin ↑ | Thick ↓ | Leak ↓ |
> |---|---|---:|---:|---:|---:|---:|---:|
> | KITTI | SplatAD | 17.14 | 0.521 | 59.2 | 1.55 | 1.38 | 22.1 |
> | KITTI | EnerGS+SplatAD | 18.32 | 0.545 | 37.6 | 1.68 | 1.44 | 7.0 |
> | Waymo | SplatAD | 26.20 | 0.787 | 44.3 | 3.27 | 0.73 | 1.4 |
> | Waymo | EnerGS+SplatAD | 26.34 | 0.784 | 44.5 | 3.22 | 0.73 | 1.2 |
>
> These results show that EnerGS remains effective when combined with a LiDAR-prior-based pipeline. On KITTI, where the LiDAR points are sparser, it consistently improves both photometric quality and geometric regularity, with a notably lower Leak score. On Waymo, the gains are smaller, but the overall performance remains comparable while still reducing leakage artifacts. This is consistent with our motivation: EnerGS is particularly beneficial when the geometric prior is partial or sparse, while remaining compatible with stronger LiDAR-based priors. We will include this comparison in the revised paper.

---

> > ### Author Rebuttal · Reviewer_8wdk · 2026-04-03
> >
> > Thanks for your response!
> >
> > I still have 2 questions:
> >
> > 1: **Sensitivity to Hyperparameters**
> >
> > The framework introduces several new hyperparameters (w_occ, σ_occ, w_unk, σ_unk, λ_free, δ, τ). While defaults are reported in Table 4, the paper provides limited sensitivity analysis. It remains unclear how robustly the method performs across scenes with very different LiDAR densities, sensor configurations, or urban layouts. A dedicated hyperparameter sensitivity study would strengthen the empirical claims considerably.
> >
> > 2: *Limitation of the MAP Formulation*
> >
> > The MAP formulation (Eq. 4) assumes the geometric prior is independent of the photometric appearance. We treat the LiDAR measurements not just as a set of points, but as a source of a spatial probability density function. Assuming the geometric prior is independent of the photometric appearance, the objective is to maximize the posterior. This conditional independence assumption, while practically convenient, is not always valid, in photometrically ambiguous regions (e.g., specular surfaces, featureless walls), geometry and appearance are often entangled. The implications of this assumption violation are not discussed.

---

> > > ### Author Response · Authors · 2026-04-06
> > >
> > > Thank you for the feedback! We hope that our previous response has fully addressed your concerns and questions in the first rebuttal round. Below are the answers to the additional questions:
> > >
> > > **Sensitivity to Hyperparameters (Q4)**
> > >
> > > Overall, we evaluated hyperparameter sensitivity across all core components of EnerGS and found the method to be broadly stable: most perturbations cause only negligible changes, while the only clear degradation arises in extreme settings that explicitly violate the intended OCC/UNK design principle.
> > >
> > > (1) Clarification of parameter coupling.
> > >
> > > The attractive energy gradients share the same form: $$\nabla E =\frac{w}{\sigma^2}\cdot d \cdot \exp\left(-\frac{d^2}{2\sigma^2}\right)\nabla d.$$
> > >
> > > In the near-surface regime ($d \ll \sigma$), $$\exp\left(-\frac{d^2}{2\sigma^2}\right) \approx 1,$$
> > > So the force becomes: $$F \approx \alpha \cdot d, \qquad \alpha = \frac{w}{\sigma^2},$$
> > >
> > > where $\alpha$ is the effective spring constant. To test this approximation, we vary $\sigma$ by $8\times$ while fixing $\alpha$, and observe only small changes on both datasets: OCC gives an averaged PSNR std/range of 0.04 / 0.10 dB, and UNK 0.05 / 0.12 dB. This supports $F \approx \alpha d$ with $\alpha = w/\sigma^2$: since most Gaussians converge near target surfaces ($d \ll \sigma$), the effective stiffness $\alpha$, not the far-field potential shape, dominates. The 7 nominal hyperparameters thus reduce to 5 independent ones.
> > >
> > > (2) Sensitivity to parameter perturbation.
> > >
> > > We sweep 6 parameters individually (all others fixed). Note that the sensitivity of $\tau$ has been addressed in Q2 from the reviewer 2XBB.
> > >
> > > |Parameter|Range| PSNR Std|PSNR Range|SSIM Std|
> > > |---:|---:|---:|---:|---:|
> > > | $w_{unk}$|[0, 1.0]|0.048|0.135|0.0018|
> > > | $\sigma_{occ}$|[0.25, 4.0]|0.055|0.138|0.0016|
> > > | $\lambda_{free}$|[0.25, 4.0]|0.060|0.178|0.0019|
> > > | $\delta$|[0.1, 2.0]|0.062|0.165|0.0017|
> > > | $w_{occ}$|[0.25, 4.0]|0.065|0.183|0.0025|
> > > | $\sigma_{unk}$|[0.5, 8.0]|0.383|1.111|0.0135|
> > >
> > > The large variance of $\sigma_{unk}$ comes from the extreme case $\sigma_{unk} = 0.5$, which increases $\alpha_{unk}$ from 0.0625 to 1.0 (a $16\times$ increase in the UNK effective force). This follows from the $w/\sigma^2$ coupling in (1), rather than an independent sensitivity to $\sigma_{unk}$ itself. Except for this extreme $\sigma_{unk}$ setting, all parameters show PSNR Std below 0.07 dB, well within the stochastic training noise floor.
> > >
> > > (3) Different LiDAR configurations.
> > >
> > > Our experiments span Waymo and KITTI, which differ fundamentally in LiDAR density, sensor configuration, and urban layout. Waymo uses denser LiDAR and multiple cameras across several U.S. cities, while KITTI uses sparser LiDAR and a monocular camera in Karlsruhe, Germany. Their sensitivity profiles are broadly consistent: excluding the extreme $\sigma_{unk}$ case in (2), all parameters stay below 0.1 dB PSNR Std on both datasets. Minor differences reflect the expected physical asymmetry: Waymo is slightly more sensitive to UNK parameters, and KITTI to OCC parameters, though both remain within the noise floor.
> > >
> > > (4) Extreme OCC/UNK ratios for the validation of the design principle.
> > >
> > > A natural question is whether the relative strength between OCC and UNK attraction requires tuning. We test this by fixing the geometric mean $$\sqrt{\alpha_{occ} \cdot \alpha_{unk}} = 0.25$$ and sweeping the ratio $$R = \frac{\alpha_{occ}}{\alpha_{unk}}$$ from UNK- to OCC-dominant:
> > >
> > > |R|Interpretation |Waymo $\Delta$PSNR|KITTI $\Delta$PSNR|
> > > |---:|---:|---:|---:|
> > > |1/16|UNK $\gg$ OCC|-0.66|-0.23|
> > > |1/4|UNK $>$ OCC|-0.29|-0.10|
> > > |4|OCC $>$ UNK|-0.05|-0.07|
> > > |16|OCC $\gg$ UNK|0.00|0.00|
> > > |64|OCC dominant|-0.05|-0.01|
> > >
> > > The negative control succeeds: violating the physical prior ($\alpha_{unk} \gg \alpha_{occ}$) degrades PSNR consistently on both datasets, confirming that OCC surfaces should attract more strongly than UNK regions.
> > >
> > > **Assumption in MAP Formulation (Q5)**
> > >
> > > Eq. (4) assumes conditional independence at the sensor-observation level: given the scene parameters $\Theta$, RGB and LiDAR observations are modeled as independent sensing processes. This reflects that camera and LiDAR operate through different physical mechanisms, standard in LiDAR-RGB fusion. We believe the reviewer is pointing to a related but different issue: not a failure of this conditional independence assumption, but the ambiguity of disentangling geometry from appearance in specular or textureless regions. This is precisely where LiDAR provides complementary geometric constraints, a key motivation for LiDAR-based 3DGS. In EnerGS, such cases are further handled by robust occupied-space modeling, weak priors in $\Omega_{unk}$, and decoupled optimization, under which unsupported Gaussians are naturally suppressed or pruned. Therefore, we do not believe this constitutes a violation of the assumption in Eq. (4); rather, it reflects a challenging multimodal ambiguity that our design explicitly mitigates. We will clarify this in the revision.

---

### Decision · Program_Chairs · 2026-04-30

**Decision:**

Accept (regular)

**Comment:**

This submission eventually got three positive recommendations and one negative recommendation. Initially, the reviewers were concerned about the runtime efficiency and complexity, the evaluation, and the technical design. The authors managed to address most of these concerns in the rebuttal. During the discussion among the authors and the reviewers, most of the reviewers confirmed that their concerns had been fully addressed. While the reviewer who gave negative recommendations did not respond to the authors' reply and several AC’s notification. Thus, all reviewers did not reach a consensus. The AC read through the manuscript, all reviews, the rebuttal, the discussions among the authors and the reviewers, and the author AC confidential comment, the AC agreed with most of the reviewers, and liked the idea of the paper. The AC also confirmed that the authors’ response basically addressed the concerns of the reviewer who did not provide feedback. Per these, the AC made a decision of acceptance. This decision was approved by the SAC as well.